Resource

# The interactome of KRAB zinc finger proteins reveals the evolutionary history of their functional diversification

Pierre-Yves Helleboid[1,†], Moritz Heusel[2,†], Julien Duc[1], Cécile Piot[1], Christian W Thorball[1], Andrea Coluccio[1], Julien Pontis[1], Michaël Imbeault[1], Priscilla Turelli[1], Ruedi Aebersold[2,3,*] & Didier Trono[1,**]

## Abstract

Krüppel-associated box (KRAB)-containing zinc finger proteins (KZFPs) are encoded in the hundreds by the genomes of higher vertebrates, and many act with the heterochromatin-inducing KAP1 as repressors of transposable elements (TEs) during early embryogenesis. Yet, their widespread expression in adult tissues and enrichment at other genetic loci indicate additional roles. Here, we characterized the protein interactome of 101 of the ~350 human KZFPs. Consistent with their targeting of TEs, most KZFPs conserved up to placental mammals essentially recruit KAP1 and associated effectors. In contrast, a subset of more ancient KZFPs rather interacts with factors related to functions such as genome architecture or RNA processing. Nevertheless, KZFPs from coelacanth, our most distant KZFP-encoding relative, bind the cognate KAP1. These results support a hypothetical model whereby KZFPs first emerged as TE-controlling repressors, were continuously renewed by turnover of their hosts' TE loads, and occasionally produced derivatives that escaped this evolutionary flushing by development and exaptation of novel functions.

**Keywords** coelacanth; evolution; KAP1; KRAB zinc finger proteins; mass spectrometry
**Subject Categories** Post-translational Modifications & Proteolysis; Chromatin, Transcription & Genomics
**The EMBO Journal** (2019) 38: e101220

## Introduction

*KZFP* genes emerged in the last common ancestor of coelacanth (*Latimeria chalumnae*), lungfishes, and tetrapods some 413 million

years ago (MYA) (Imbeault *et al*, 2017). Their products harbor an N-terminal KRAB (Krüppel-associated box) domain related to that of Meisetz (a.k.a. PRDM9), a protein that originated prior to the divergence of chordates and echinoderms, and a C-terminal array of zinc fingers (ZNF) with sequence-specific DNA-binding potential (Urrutia, 2003; Birtle & Ponting, 2006; Imbeault *et al*, 2017). *KZFP* genes multiplied by gene and segment duplication to count today more than 350 and 700 representatives in the human and mouse genomes, respectively (Urrutia, 2003; Kauzlaric *et al*, 2017). A majority of human KZFPs including all primate-restricted family members target sequences derived from TEs, that is, DNA transposons, ERVs (endogenous retroviruses), LINEs, SINEs (long and short interspersed nuclear elements, respectively), or SVAs (SINE-variable region-Alu) (Schmitges *et al*, 2016; Imbeault *et al*, 2017). However, more ancient family members do not bind recognizable TEs but are rather found at promoters or over gene bodies (Frietze *et al*, 2010a,b; Imbeault *et al*, 2017).

The KRAB domain was initially characterized as capable of recruiting KAP1 (KRAB-associated protein 1), a tripartite-motif (TRIM) protein that serves as a scaffold for a heterochromatin-inducing complex comprising notably the histone 3 lysine 9 methyltransferase SETDB1, HP1 (heterochromatin protein 1), and the histone deacetylase-containing NuRD (nucleosome remodeling deacetylase) complex (Friedman *et al*, 1996; Ryan *et al*, 1999; Schultz *et al*, 2001, 2002). Accordingly, many KZFPs act in association with KAP1 and associated effectors to repress TEs during the genomic reprogramming that takes place during the earliest stages of embryogenesis (Wolf & Goff, 2009; Matsui *et al*, 2010; Rowe *et al*, 2010; Castro-Diaz *et al*, 2014). However, KAP1 is bound neither by human PRDM9 nor by several other highly conserved KZFPs harboring additional N-terminal domains such as SCAN, which can promote oligomerization, or DUF3669, a region of still elusive function, suggesting that KAP1 binding and repressor activity are recently evolved properties of KZFPs (Okumura *et al*, 1997; Williams *et al*,

---
1 School of Life Sciences, Ecole Polytechnique Fédérale de Lausanne, Lausanne, Switzerland
2 Department of Biology, Institute of Molecular Systems Biology, ETH Zurich, Zurich, Switzerland
3 Faculty of Science, University of Zurich, Zurich, Switzerland
*Corresponding author. Tel: +41 446 333170; E-mail: aebersold@imsb.biol.ethz.ch
**Corresponding author. Tel: +41 216 931761; E-mail: didier.trono@epfl.ch
† These authors contributed equally to this work

1999; Schumacher *et al*, 2000; Birtle & Ponting, 2006; Itokawa *et al*, 2009; Liu *et al*, 2014; Patel *et al*, 2016; Imai *et al*, 2017). Moreover, *KZFP* genes display broad and diverse patterns of expression and have been linked to biological events such as genomic imprinting, RNA metabolism, cell differentiation, metabolic control, and meiotic recombination (Wagner *et al*, 2000; Hayashi & Matsui, 2006; Quenneville *et al*, 2011; Zeng *et al*, 2012; Lupo *et al*, 2013; Ecco *et al*, 2017; Yang *et al*, 2017). How these other effects are accomplished is partly unknown, but they suggest that KZFPs associate with a range of cofactors extending well beyond the sole inducers of transcriptional repression. Undertaken to explore this complexity, the present study reveals the breadth and evolutionary history of the functional diversification of KZFPs.

## Results

We selected 101 human KZFPs over a range of evolutionary ages, domain compositions, and genomic targets so as to constitute a sample representative of the whole family. Using 293T cell lines overexpressing HA-tagged versions of these proteins (Imbeault *et al*, 2017), we first determined their subcellular localization by indirect immunofluorescence (IF) microscopy (Table EV1). A majority of these KZFPs were almost exclusively nuclear, as illustrated for ZNF93, but some displayed unusual sub-nuclear or predominantly cytoplasmic localizations, such as the nucleolus-enriched ZNF79 or the endoplasmic reticulum (ER)-associated ZNF546 (Fig 1A). We then set out to define the protein interactome of these KZFPs by affinity purification and mass spectrometry (AP-MS) essentially as previously described (Varjosalo *et al*, 2013) (Fig EV1A) with modifications aimed at optimizing the workflow for the analysis of DNA-associated proteins with high sensitivity. To narrow down identified proteins into a high-quality list of interactors, we used the significance analysis of interactome (Choi *et al*, 2011), (SAINT)-calculated false discovery rate (FDR), and spectral counts fold enrichment as well as a subcellular localization filter using our IF data for the baits and the Human Protein Atlas annotations for their preys (Thul *et al*, 2017). The resulting human KZFP interactome formed a high-density connectivity map of 887 high-confidence associations between the 101 baits and 219 preys displaying similar subcellular localization patterns. Confirming that our list of KZFP-binding proteins was not contaminated by loci-specific DNA-binding contaminants, KZFPs with partly overlapping genomic targets (Imbeault *et al*, 2017) did not share more partners than random pairs of these proteins (Fig EV1B). In order to confirm the validity of our dataset, we compared our results with the BioGRID protein–protein interaction dataset (Stark, 2005) (Table EV2). We found that 88 (about 8%) of the interactions documented in our system had been previously detected through either AP-MS or other approaches such as two-hybrid screens (Fig EV1C).

We used our AP-MS results to build a KZFP interactome global network (Fig 1B and Appendix Fig S1). It was centered on KAP1 (Fig 1C) and displayed KZFPs associating with specific preys at its periphery (Fig 1D and E), as well as three KZFPs with no detected interactors (Fig 1F). Its core encompassed the majority of the KZFP baits and their most frequent interactor, KAP1, which was frequently detected together with HP1α and HP1γ, the deacetylase SIRT1, and the ATP-dependent helicase SMARCAD1, all previously

identified as KAP1 interactors (Ryan *et al*, 1999; Rowbotham *et al*, 2011; Lin *et al*, 2015) (Fig 1C). SIRT1 associated with the KAP1-interacting ZNF138 and ZNF793 but not with ZKSCAN3, a weak recruiter of the corepressor (Fig EV2A), consistent with such KAP1-mediated interaction. Additional proteins involved in post-translational modifications, such as phosphorylation and ubiquitylation, were repeatedly part of KAP1-comprising KZFP interactomes (Fig EV2B). Our analyses also identified mediators of nuclear import in association with more than half (56) of nuclear KZFPs, whether KAP1 was (karyopherin β, the karyopherins α KPNB1 and KPNA2) or not (karyopherin β-like importins IPO7 and IPO8) systematically present (Fig EV2B and C).

In line with the known dimerization potential of SCAN (Williams *et al*, 1999), another sub-network emerged that was based on associations between proteins harboring this domain (Fig 2A). Nine out of 17 SCAN-KZFPs were found in such complexes, which for six of them also included the non-KRAB SCAN-containing ZFP ZNF24, thought to play a role in transcription and DNA replication (Jia *et al*, 2013; Lopez-Contreras *et al*, 2013). These results concur with some of the putative SCAN-mediated interactions noted in previous studies (Huttlin *et al*, 2015; Schmitges *et al*, 2016).

The capture of ZNF282 by ZNF398 attracted our attention, as both of these KZFPs contain a DUF3669 domain, and interactions between DUF3669-containing KZFPs were previously recorded (Gao *et al*, 2008; Kang & Shin, 2015; Huttlin *et al*, 2017) (Appendix Fig S2). Confirming the ability of DUF3669 to mediate protein–protein interactions, we could co-immunoprecipitate HA-tagged ZNF282 with full length but not DUF3669-deleted version of GFP-tagged ZNF398 (Fig 2B), and we could pull-down the ZNF398 DUF3669 domain with the corresponding fragment of ZNF282, but not with its KRAB counterpart (Fig 2C). We thus conclude that the DUF3669 domain can trigger associations between KZFPs, although *in vitro* experiments with purified proteins will be needed to confirm that it is through a direct interaction.

Outside of the KAP1-centered network core, we delineated other interactomes displaying uncommon associations (Fig 1D and E). First, we focused on preys captured by only two to four KZFPs (Fig 3A). Those found together with KAP1 related to nucleosome formation and modification or involved other TRIM family proteins or cytoplasmic chaperones sharing subcellular localizations with their baits as for ZNF546, ZNF304, and ZNF283 (Fig 1A, Appendix Fig S3A). Several subunits of translation-promoting eukaryotic initiation factor 3 (eIF3) were also part of the interactomes of the weak KAP1 recruiters ZNF79, ZKSCAN4, and ZNF746.

We then focused on the 16 KZFPs interacting with three or more unique preys, i.e., preys that were detected in association with only one KZFP, reasoning that their interactors would provide more detailed hints on their biological roles (Fig 3B). Of note, 15 interactions between a KZFP and a unique interactor were already observed in published datasets (Table EV2). Several of these unique preys pointed to RNA-related functions such as unique partners detected with the eIF3-associated ZKSCAN4 and ZNF746 or the RNA-processing paraspeckle-forming proteins PSPC1, NONO, and SFPQ for ZNF213, corroborated by their similar sub-nuclear localization patterns in these structures (Hata *et al*, 2008) (Appendix Fig S3B). Furthermore, factors involved in cell-cycle regulation (cyclin and associated kinases) bound ZNF20, and components of the chromosome-organizing SMC complexes associated with ZNF597 and

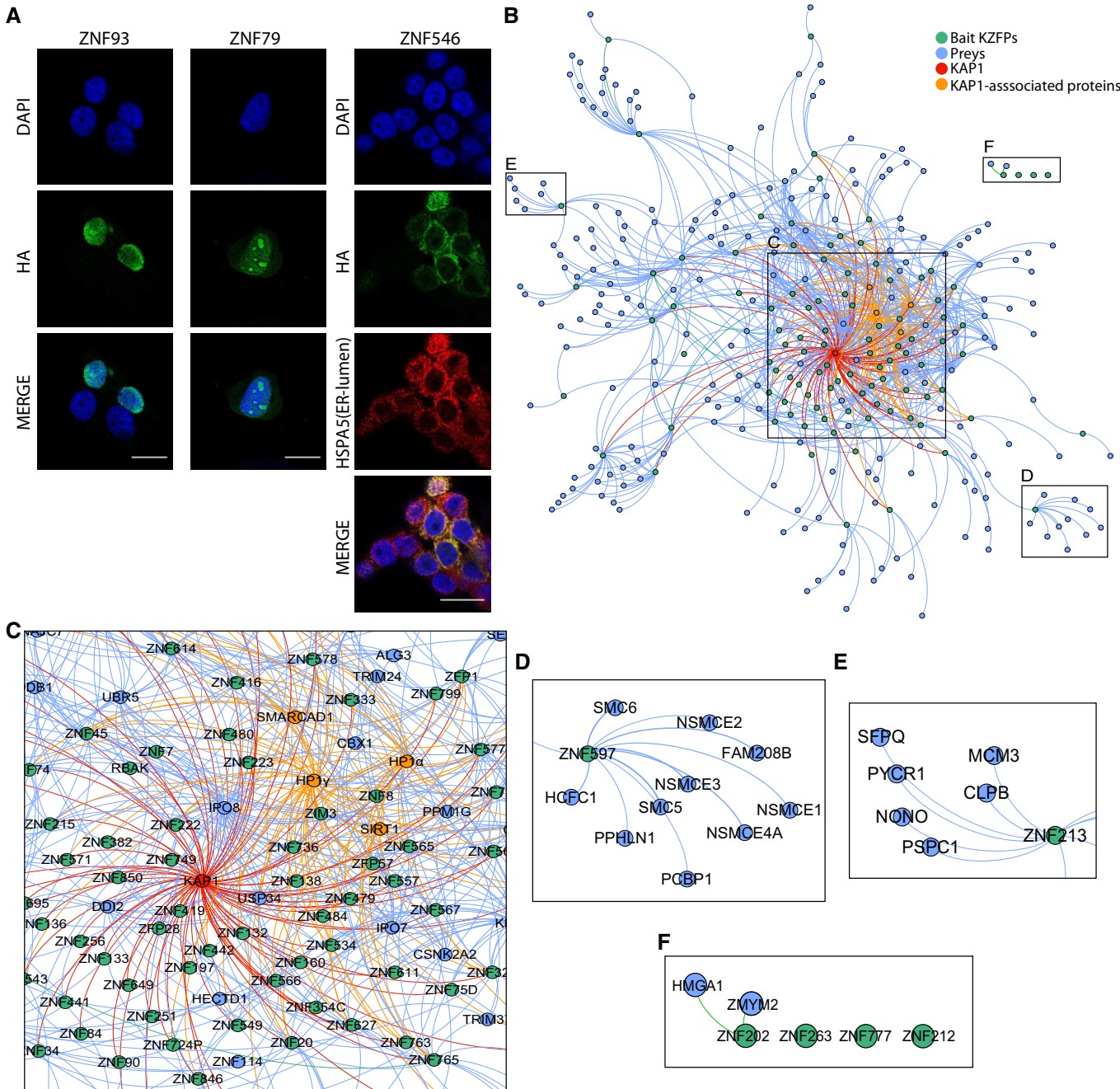

**Figure 1. KZFP IFs and interactome.**

A  IF by confocal microscopy performed on KRAB-HA-overexpressing 293T cells. Staining was performed with anti-HA (Alexa-488, green) and anti-HSPA5 (Alexa-647, red) antibodies, and DNA was stained with DAPI (blue). The scale bar represents 20 μm.

B  Force-directed network representing high-confidence interactions. Each bait KZFP is represented by a blue circle ("node") and linked to green nodes that represent its preys. KAP1 is represented in red and its associated proteins SIRT1, SMARCAD1, and HP1α and HP1γ in orange. All interactions represented are below the false discovery rate of 1%. The topology of the network is established by a force-directed process that follows certain rules: All nodes repel each other and are attracted to the center by artificial "gravity", and nodes with links attract each other. Weighted links between any nodes are based on the average fold change over controls.

C  Zoom-in on the KAP1-centered core of the interactome.

D, E  Zoom-in on KZFPs enriched with unique interactors.

F  Zoom-in on the subset of KZFPs not connected to the main interactome.

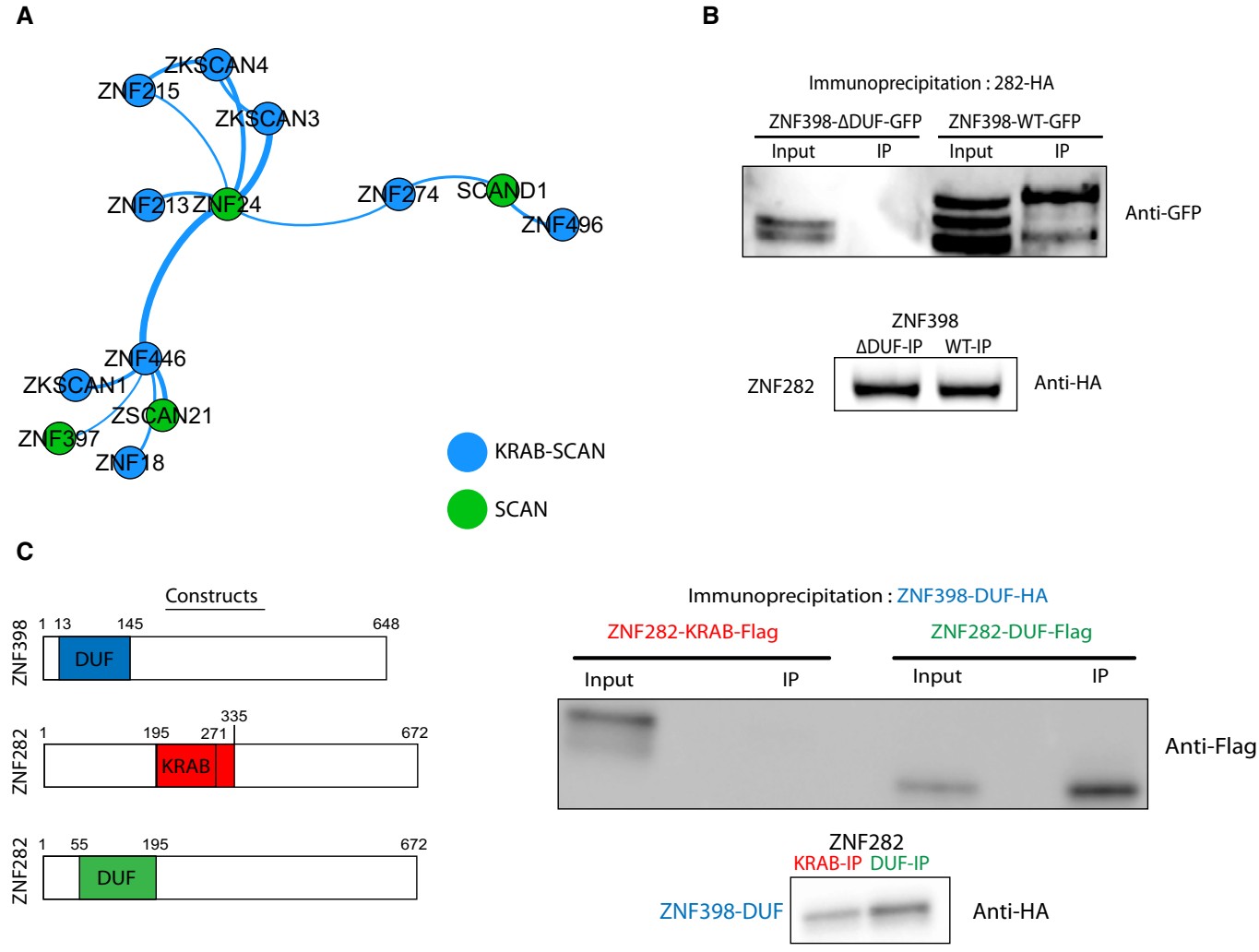

**Figure 2. SCAN and DUF3669 domains are involved in oligomerization.**

A  Force-directed network representing the SCAN interactome displaying SCAN-containing baits and their SCAN-containing preys.

B  HA immunoprecipitation of stably expressed ZNF282-HA in cells previously transfected with ΔDUF3669-ZNF398-GFP and WT-ZNF398-GFP followed by the detection of ZNF398 constructs in the IPs through Western blot using an anti-GFP antibody. Bottom: Western blot using an anti-HA antibody on the IPs. Input = cellular lysate, IP = immunoprecipitate.

C  Left: DUF3669-only and KRAB domain constructs used. Right: HA immunoprecipitation in cells previously co-transfected with ZNF398-DUF3669-HA and either ZNF282-KRAB-Flag or ZNF282-DUF3669-Flag followed by the detection of either of these protein constructs in the IPs through Western blot using an anti-Flag antibody. Bottom: Western blot using an anti-HA antibody on the IPs.

Source data are available online for this figure.

ZNF3. Finally, TFIIIC subunits were part of the interactome of ZNF764, previously found to cooperate with CTCF in establishing genomic boundaries (Moqtaderi *et al*, 2010). Correspondingly, chromatin immunoprecipitation/deep sequencing (ChIP-seq) studies found the three proteins co-localized significantly on the genome (Fig 3C).

We further documented the genomic recruitment of the 16 KZFPs interacting with 3 or more unique preys, completing a previously established dataset (Imbeault *et al*, 2017) with additional ChIP-seq studies (Table EV3). A majority was found at TEs or gene transcription start sites (TSS) (Fig 3D), but some were rather associated with other entities such as imprinting control regions (ICRs) for ZNF445 (Takahashi *et al*, 2019) and CTCF binding sites for ZNF764. In addition, few or no ChIP-seq peaks were detected for ZNF446, ZNF546, ZNF213, and ZNF597, indicating different modalities or absence of DNA binding.

Except for the MER51A/E-binding ZNF20, TE-binding KZFPs displaying unique interactomes were enriched over LINEs, which often recruited several of them (Imbeault *et al*, 2017) (Appendix Fig S4). For instance, ZNF93, ZNF765, and ZNF248 bound to L1PA6 and L1PA5 LINE1 integrants, for the first two over their 5′ untranslated and for the third over their ORF2-coding regions (Fig 3E). The presence of escape mutations in the LINE1 lineage (Jacobs *et al*, 2014; Imbeault *et al*, 2017) indicates that ZNF93, ZNF765, and ZNF248 initially acted as *bona fide* inhibitors of transposition. Yet, their persistent association with L1PA5/PA6 integrants suggests that

**A**

## Rare interactors (present in more than 1 and less than 5 interactomes)

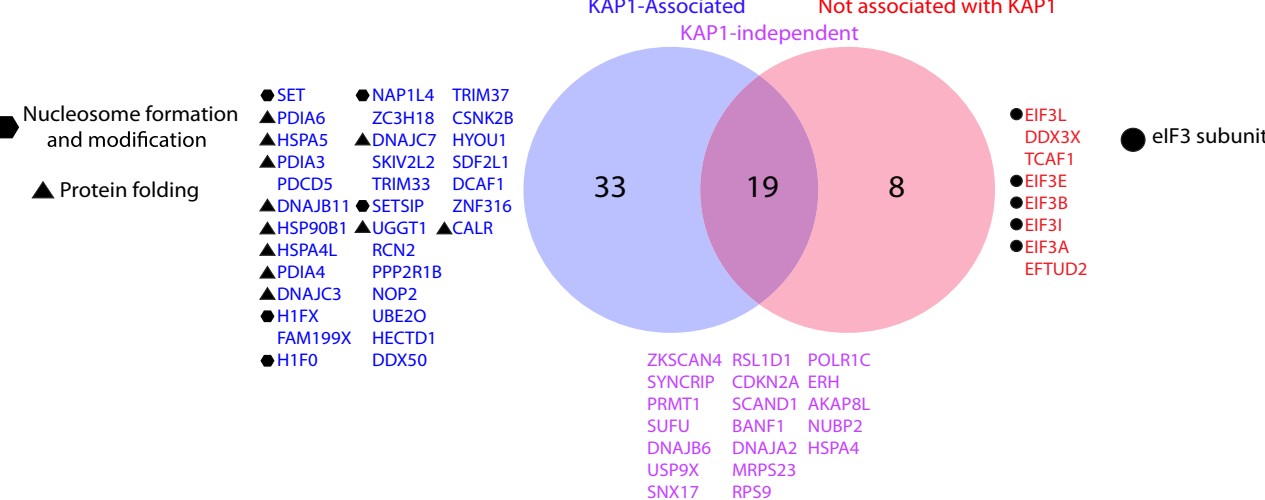

**B**

| KZFP | N-term domain | Unique interactors | Associated features |
|---|---|---|---|
| ZNF446 | SCAN | MAGED1,SEC16A,ZNF18,DNAJA1,DNAJA3, ZSCAN12, TUBB4A,HSPB1,ZKSCAN1, SCRIB,ZNF397,RFWD2,LZTS2, ZSCAN21, TRAP1,RPS27 | SCAN-containing proteins |
| ZNF597 | - | HCFC1,FAM208B,SMC5,PPHLN1,NSMCE1, NSMCE2, NSMEC3,SMC6,NSMCE4A,PCBP1 | SMC5/6 complex |
| ZNF3 | - | LONP1,UTRN,TARS2,SMC1A,PYCR3,SMC2,GTF2I | Condensin/Cohesin |
| ZNF213 | SCAN | CLPB,PSPC1,NONO,PYCR1,MCM3,SFPQ | Paraspeckles |
| ZNF764 | - | GTF3C1,GTF3C2,GTF3C3,GTF3C4,GTF3C5 | Transcription factor III C |
| ZFP1 | - | TXLNA,TNPO1,MAP3K20,PRPF19 | - |
| ZNF546 | - | CALU,RCN2,HNRNPH1,GANAB | Endoplasmic reticulum lumen |
| ZNF445 | SCAN | WDR62,HIST1H1C,NUMA1,SNRPD2 | - |
| ZNF248 | - | CCT4, CCT5, CCT8, TCP1 | Tric chaperonne |
| ZNF2 | - | COPG1,COPA,RPS27A | Coatamers |
| ZNF20 | - | CCNA2,CDK2,CDK1 | Cyclin/CDK |
| ZNF765 | - | ZNF460,ZNF677,c18orf25 | Zinc finger proteins |
| ZNF846 | - | RNH1,SART3,RBBP7 | - |
| ZNF93 | - | MRPS31,HIST1H1E,RPL21 | - |
| ZKSCAN4 | SCAN | RAD50,SNRNP200,DHX15 | RNA helicases |
| ZNF746 | DUF3669 | EIF3F,TPP1,PRRC2B | RNA-binding proteins |

**C**

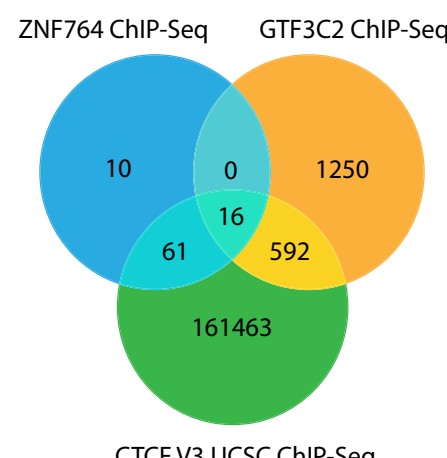

Fisher's exact test two tails p value ZNF764-GTF3C2 : 5.1429e-38
Fisher's exact test two tails p value ZNF764-CTCF : 6.75e-95

**D**

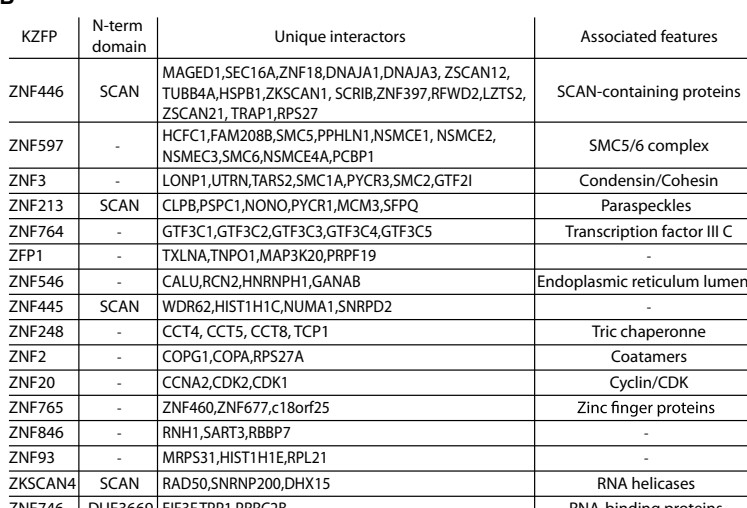

**E**

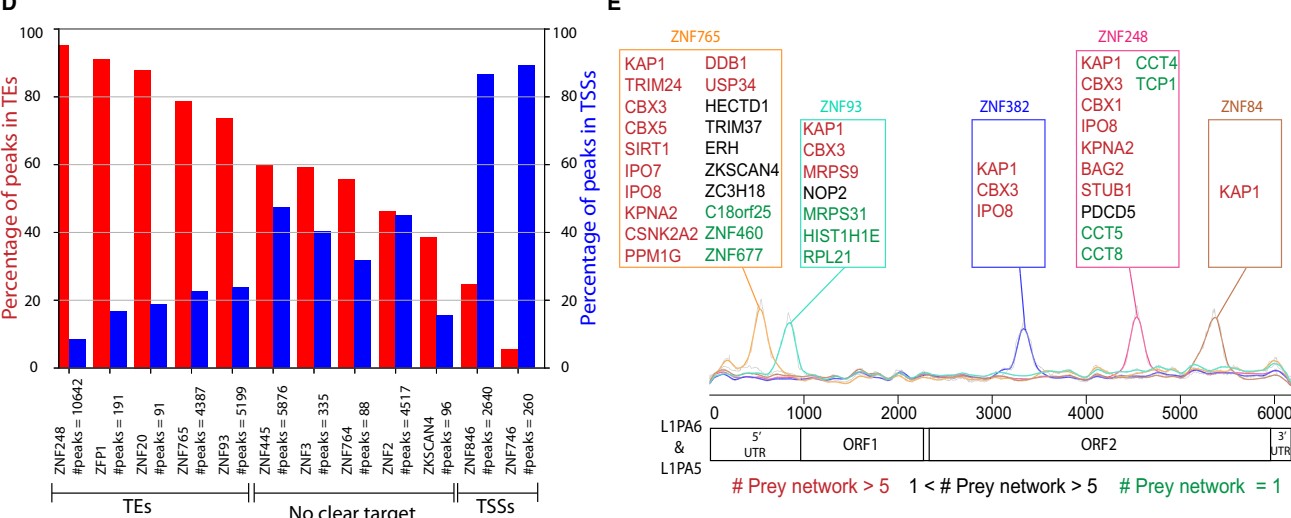

**Figure 3.**

**Figure 3. A sub-population of KZFPs displays rare and unique interactors.**

A   Venn diagram representing all preys detected in less than 5 and more than 1 KZFP interactomes. In blue, are shown the preys that only appear in interactomes alongside KAP1, in red the preys that only appear in interactomes devoid of KAP1, and in purple the preys that are in both types of interactomes.

B   This table displays all the KZFPs associating with 3 or more unique interactors, the identity of these interactors as well as the features associated with them.

C   Venn diagram representing the binding sites overlaps between ZNF764, TFIIIC subunit GTF3C2, and CTCF. The GTF3C2 bedfile was obtained from Encode (ENCFF002CYL), and CTCF bedfile was obtained from Encode version 3. The overlap and resulting *P*-values were obtained using the Bedtools Fisher exact test.

D   Histograms representing the percentage of KZFPs (identified in Fig 3B) binding sites falling in TEs (red) or TSSs (blue).

E   KZFPs binding sites on L1PA5 and L1PA6. Top: Boxes containing the interactors of KZFPs found enriched on L1PA5 and L1PA6. Middle: plot showing the average ChIP-exo signals (scaled between 0 and 1) for each selected KZFP plotted on top of L1PA5 and L1PA6 multiple sequence alignment (MSA). Bottom: schematic representation of L1PA5 and L1PA6 different domains.

these KZFPs, while perhaps still repressing transcription, do so no longer to block retrotransposition, since their target retroelements have long lost all spreading potential. ZNF765 is recruited over the L1PA5/PA6 promoter region, and two of its interactors stand out as potential regulators of LINE1 transcripts: the RNA-binding proteins interactor ZKSCAN4 and ZC3H18, known to be involved in RNA export, anabolism, and catabolism (Chi *et al*, 2014; Winczura *et al*, 2018). Furthermore, L1PA6 and L1PA5 integrants also recruit over their 3′ half two KZFPs with conventional KAP1-centered interactomes, ZNF382 and ZNF84, for which there is no sign of mutational escape, strongly suggesting that these KZFPs, in spite of their repressor potential, never limited the spread of these retrotransposons.

We then turned our attention toward KAP1, the most common partner of KZFPs. Interestingly, our network reflected a range of affinities between individual KZFPs and the corepressor (Fig 4A), translating in differential KAP1 recruitment strengths (KAP1FC). With an arbitrary cut-off at three, this parameter derived from spectral counts enrichment delineated two groups of KZFPs, which we qualified as strong and weak KAP1 binders (Figs 4A and EV3A and B). Upon examining which KZFP features correlated with KAP1 recruitment, we first noticed that family members with < 40% of ChIP-seq peaks on TEs interacted on average less strongly with KAP1 (Fig 4B). The KRAB domain typically comprises an obligatory A-box bearing the residues necessary for KAP1 recruitment and a facultative B-box (Urrutia, 2003). B-box displaying KZFPs predominantly yielded high KAP1FC values, whereas their B-box-less counterparts were split among strong and weak KAP1 binders (Fig EV3C), confirming the enhancing but non-essential role of this subdomain (Vissing *et al*, 1995). We built a phylogenetic tree based on the KRAB-A-boxes of 346 protein-coding human KZFPs (Fig 4C). A majority clustered as a homogeneous group harboring very few

amino acid differences, and we termed these standard-KRAB KZFPs (sKZFPs). A heterogeneous set of variant KRAB (vK) emerged that displayed a very significant degree of divergence from the KRAB A-box consensus sequence. We identified such 35 vKZFPs in the human proteome (in addition to two KZFPs, ZNF333 and ZFP28, harboring both a standard and variant KRAB). Only one out of the 18 vKZFPs subjected here to AP-MS had a KAP1FC value above three, whereas the opposite was observed for 81 out of 83 tested sKZFPs (Fig 4D). By assessing their evolutionary ages (Imbeault *et al*, 2017), we determined that among tested KZFPs, those harboring a vK domain segregated in oldest age bins, conserved from placental mammals (105 MY) up to sauropsids (320 MY), whereas all but two of their sK-containing counterparts were 105 MY Old (MYO) or younger (Figs 4D and EV3D). We then confirmed that all sKZFPs, including the ones not tested in our study, presented significantly higher evolutionary ages (Fig EV3E). Older age also correlated with unusual interactomes: 13 out of 16 KZFPs interacting with more than three unique interactors were 105 MYO or older, and KZFPs displaying no interactors were all older than 105 MY (Figs 4E and 1F). Moreover, we detected less than four interactors (whereas the average number for the 101 KZFPs was 8.8) for more than half of these ancient KZFPs. For some, this might have been due to lack of expression of their functional partners in 293T cells, although these KZFPs themselves did not display abnormally low levels in these cells (Appendix Fig S5A). In contrast, more than half of the KZFPs associated with more than three unique interactors segregated in a small cluster characterized by a higher expression in testis and blood according to the GTEx database (The GTEx consortium, 2013) (Appendix Fig S5B). In sum, the most conserved KZFPs were weak KAP1 binders and displayed functionally diversified or partner-depleted interactomes indicative of non-canonical roles.

**Figure 4. Old KZFPs display an unusual and conserved KRAB domain devoid of KAP1 binding.**

A   KAP1 force-directed network. The distance KZFP-KAP1 is inversely proportional to the measured KAP1 enrichment over controls (KAP1FC). KZFPs with KAP1FC values above three were colored in blue and the ones below in green.

B   Boxplot representing the KAP1FC value of our baits in function of the percentage of their binding sites falling in TEs. Mann–Whitney two-sided rank test.

C   Human KZFPs KRAB domain A-boxes protein phylogenetic tree associated with their corresponding amino acid sequences colored according to the Clustal Zappo color scheme (residues sharing common physicochemical properties display the same color: http://www.jal-view.org/help/html/colourSchemes/zappo.html). This figure also displays a zoom-in on the variant KRAB domain cluster (right). On this zoom-in, the tested vKZFPs, whose interactomes were defined in our study, were marked by an asterisk.

D   Box plot representing KAP1FC values in function of KZFPs age. On a superimposed swarm plot, the individual vKZFPs corresponding KAP1FC values were represented by red dots and for their sKZFPs counterparts, by blue dots. Mann–Whitney two-sided rank test.

E   Boxplot representing the number of interactors of KZFPs in function of their evolutionary age. On a superimposed swarm plot, individual KZFPs number of preys were represented by dots. When green, the interactome of this KZFP contained 3 or more unique interactors (Fig 3B).

F   Boxplot representing the average dn/ds ratios for all vK (red) and sK (blue) domains displaying the same evolutionary ages. Mann–Whitney two-sided rank test.

Data information: Boxplots are shown as median, and 25th (Q1) and 75th (Q3) percentiles. The upper whisker extends to the last data point less than Q3 + 1.5*IQR, where IQR = Q3–Q1. Similarly, the lower whisker extends to the first data point greater than Q1 − 1.5*IQR.

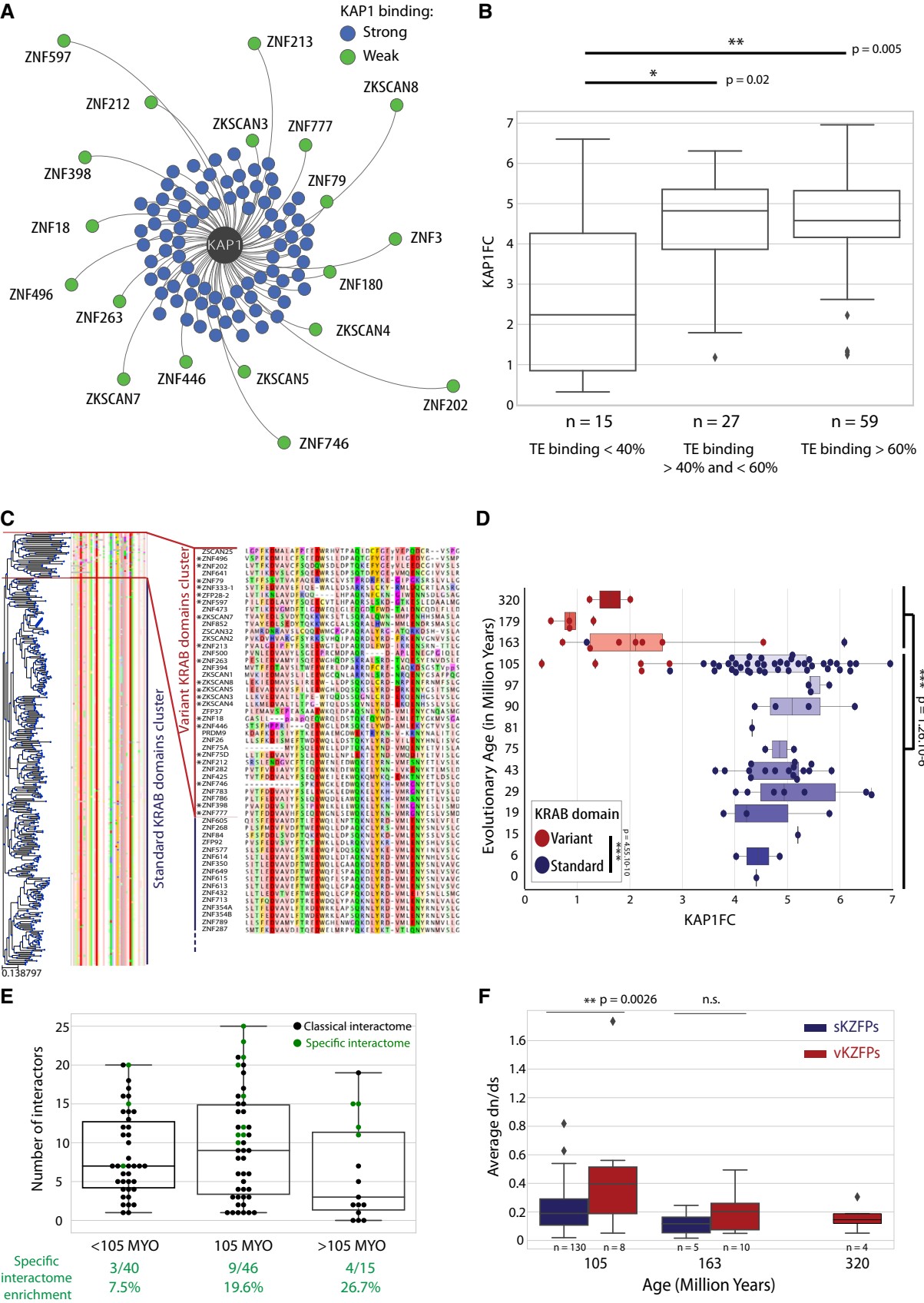

**Figure 4.**

In order to ask whether the heterogeneous vK domains were individually conserved, we calculated the ratio of non-synonymous over synonymous mutations (dn/ds) for sK and vK domains belonging to KZFPs displaying similar evolutionary ages (Fig 4F). Human vK domains conserved up to marsupials (163 MYO) exhibited dn/ds values similar to those of sK of similar ages and, as their sauropsids-conserved relatives (320 MYO), displayed medians lower than 0.3, indicating that they had been subjected to purifying selection. As well, younger vK displayed dn/ds values higher than their sK counterparts but still below the neutral selection value of one. Also, genes encoding vKZFPs presented fewer loss-of-function mutations in a collection of 123,136 exomes and 15,496 full genomes assembled in the gnomAD database (Fig EV4). Thus, although

KAP1-independent and heterogeneous in their sequences, vK domains were found to be under selective pressures indicating that they likely fulfill conserved functions.

In order to get a comprehensive representation of our data, we illustrated features such as uniqueness of interactors, KAP1 recruitment ability, subcellular localization, and KRAB domain identity on an interactors-based cluster map of our KZFPs (Fig 5). On this plot, KZFPs were clustered according to the similarity of their interactomes. KAP1-binding elements formed the central predominant unit, although the heterogeneity of their interactomes translated in subclusters revolving around additional common interactors such as IPO7, IPO8, or HP1 proteins. The three KZFPs with an IF pattern suggestive of ER localization formed a subgroup due to common

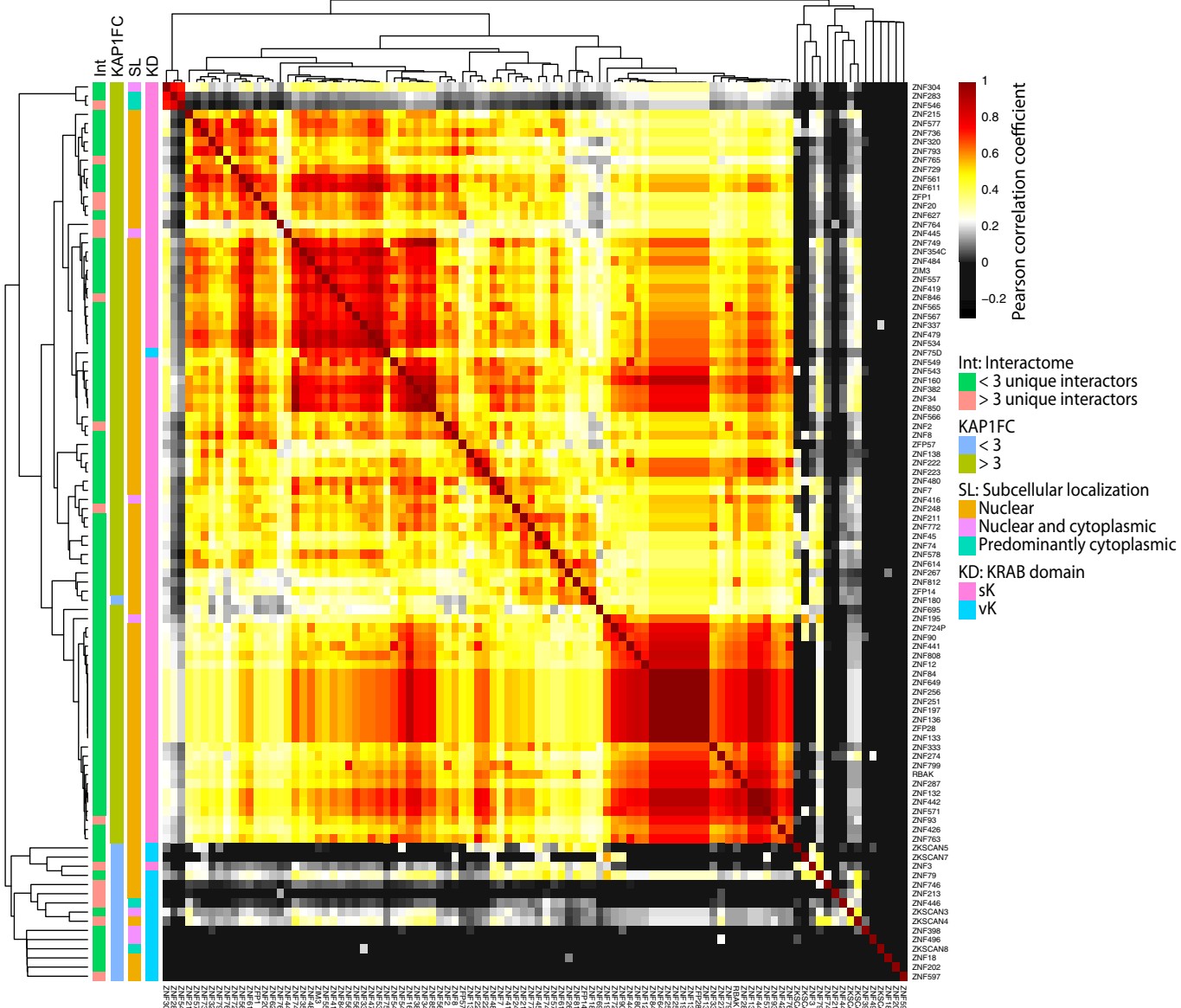

**Figure 5. Interactome-based Pearson correlation of KZFPs.**
Individual KZFP interactomes were compared by Pearson correlation analysis of prey protein fold change over control (Log2FC) values. Accordingly, KZFP bait proteins are ordered based on the similarity of their prey protein recovery profile.

ER-residing preys (Fig 5, upper left corner). Furthermore, the map was bordered by a group of KZFPs that share only few if any interactors. This subset is enriched in KZFPs endowed with unique interactors, weak KAP1 affinity, vK domains, and partial cytoplasmic localization. These data support a model whereby even though a majority of KZFPs share KAP1-related functions, older family members, notably vKZFPs, display atypical, distinct features hinting at different roles.

The most ancient human KZFPs are inefficient at recruiting KAP1, suggesting that this function evolved only long after these proteins emerged in the last common ancestor of coelacanth (*Latimeria chalumnae*) and tetrapods (this study, Birtle & Ponting,

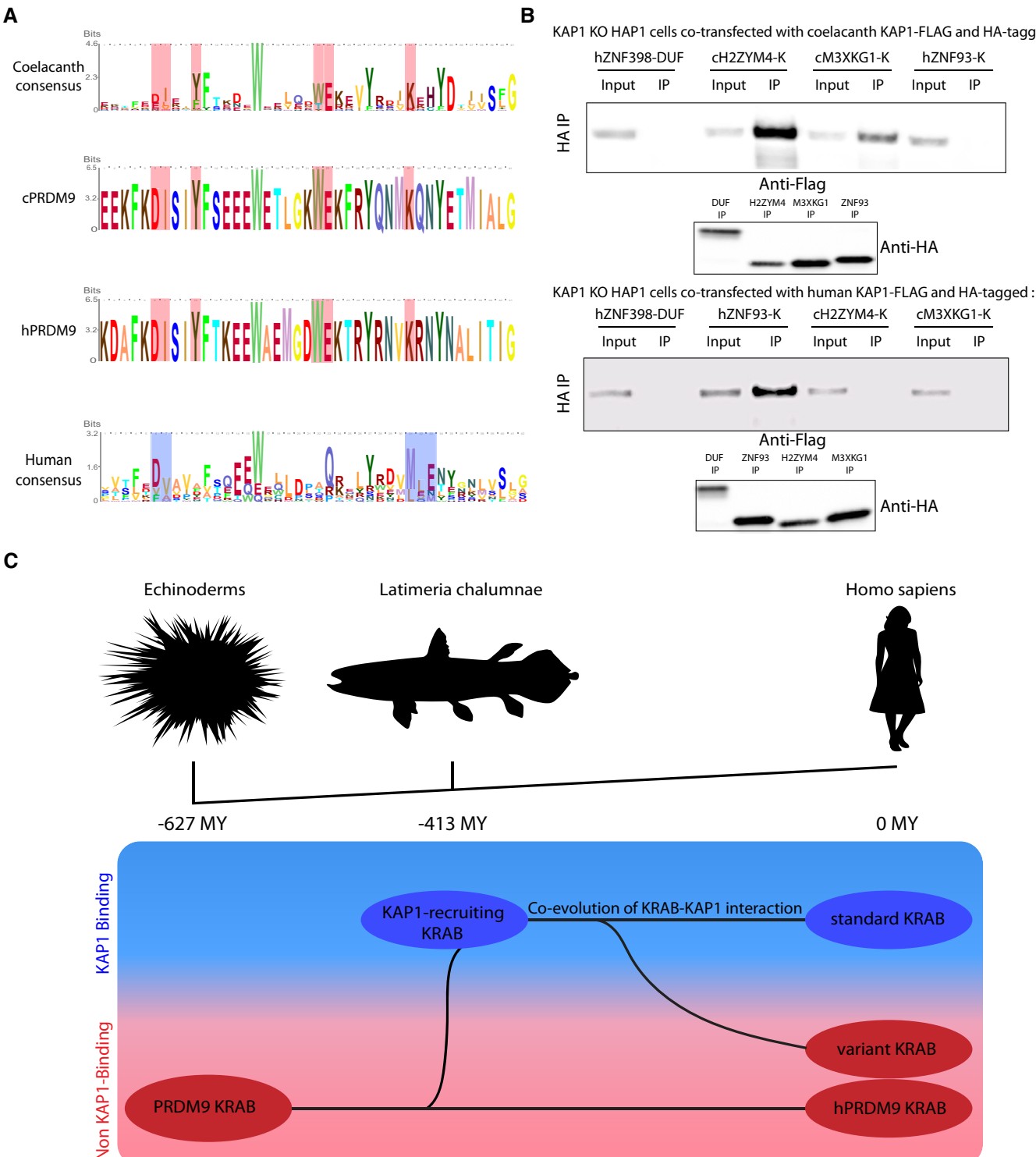

**Figure 6.**

◀

**Figure 6.  KZFPs present in the last common ancestor of coelacanth and tetrapods were KAP1 binders.**

A   Consensus sequences for KRAB A-boxes from top to bottom: coelacanth KRAB domains, coelacanth PRDM9 KRAB domain, human PRDM9 KRAB domain, and human KRAB domains. Residues conserved between the first three sequences and not present in human KRAB consensus were highlighted in red. In the human KRAB consensus, residues crucial for KAP1 recruitment were highlighted in blue (Margolin *et al*, 1994).

B   Immunoprecipitations of HA-tagged KRAB domains in order to check interaction with cKAP1. Co-transfection of Flag-tagged cKAP1 (upper panel) or Flag-tagged hKAP1 (lower panel) and HA-tagged ZNF398 DUF3669 domain negative control, H2ZYM4 and M3XKG1 cKRAB domains, and ZNF93 hKRAB domain in KAP1 KO HAP1 cells followed by HA immunoprecipitation. cKAP1 and hKAP1 presence was revealed by Western blot using an anti-Flag antibody. Input = cellular lysate, IP = immunoprecipitate. Western blot using an HA antibody on the IPs at the bottom.

C   Evolutionary KRAB model: On top, a simple phylogenetic tree represents the links between echinoderms, coelacanth, and human. At the bottom, different KRAB domain versions ranging from the ancestral PRDM9-related KRAB domain to the classical KAP1 binding human KRAB are represented. Their position on this model depends (i) under which species or clades they were detected, hence reflecting the last common ancestor in which they putatively appeared and (ii) whether they bind KAP1, in the blue rectangle, or not, in the pink rectangle. Briefly, the echinoderm-conserved PRDM9-related KRAB domain is the putative ancestor of KAP1-binding KRAB domain that emerged in the coelacanth. The KRAB domain and KAP1 co-evolved, maintaining their association in tetrapods. Meanwhile, certain KZFPs have lost KAP1 binding to become vKZFPs.

Source data are available online for this figure.

2006; Imai *et al*, 2017; Patel *et al*, 2016; Liu *et al*, 2014; Okumura *et al*, 1997; Itokawa *et al*, 2009). Supporting this hypothesis, the consensus sequence of KRAB domains of coelacanth KZFPs was closer to that of the non-KAP1 recruiter human PRDM9 than of human sKZFPs (Imai *et al*, 2017) (Figs 6A and EV5A). Still, we performed co-immunoprecipitation experiments by overexpressing coelacanth KAP1 (cKAP1) and KRAB domains (cKRAB) in *KAP1* KO human HAP1 cells to avoid interference by human KAP1 (hKAP1), which we found could oligomerize with its coelacanth ortholog (Fig EV5B). Against our expectation, the two coelacanth KRAB domains tested, derived from two KZFPs harboring KRAB sequences prototypic for this species (H2ZYM4 and M3XKG1, UniProt references), strongly associated with cKAP1. In contrast, the latter was bound by the KRAB domain neither of human ZNF93 nor of coelacanth or human PRDM9 (Fig 6B, upper panel & Fig EV5C). More predictably, considering their resemblance to the KRAB domain of PRDM9, cKRABs did not interact with human KAP1 (Fig 6B, lower panel). These results strongly suggest that KAP1 recruitment was an ancestral property of KZFPs that was maintained in distant lineages over more than 400 MY by co-evolution of the two partners of this interaction (Fig 6C). Of note, vK domains display heterogeneity in evolutionary ages, amino acid sequences, nature of N-terminal domains, and chromosomal locations that suggest several independent transition events of sKZFPs to vKZFPs.

## Discussion

This study unveils important aspects of the evolutionary history and functional diversification of human KZFPs. We started by defining the protein interactomes of 101 of these factors, selecting baits representative of the full range of conservation, domain structure, and genomic target preference displayed by this >350 member-rich family. Owing to the large number of tested baits and the lack of antibodies allowing for an efficient purification of their endogenous versions, we relied on the overexpression of tagged proteins in 293T cells. However, in addition to the use of biological replicates, SAINT probabilistic scoring, batch-specific fold-change-over-control normalization, and subcellular localization-based filters allowed us to establish a high-confidence list of interactors. This was supported by our verification that a substantial subset of the interactions detected through our approach had been previously documented in different experimental settings.

Based on our results, we propose the following hypothetical model for the evolutionary history of human KZFPs. Similar to their most modern relatives, ancestral KZFPs were endowed with KAP1 recruiting ability and had TEs in need of transcriptional control as main targets. As the pool of active TEs present in their hosts was renewed by genetic drift of resident retrotransposons (Boissinot & Furano, 2001; Khan *et al*, 2006) and successive waves of retroviral invasion (Mager & Stoye, 2015), *KZFP* genes were subjected to evolving selective pressures, leading to their turnover through replacement of older elements by new family members adapted to their hosts' changing TE loads. Our data and published studies (Porsch-Özcürümez *et al*, 2001; Shin *et al*, 2011; Chauhan *et al*, 2013) further suggest that, during the course of these events, some KZFPs evolved other functions that supplemented, modulated, or replaced transcriptional repression, a functional modification that went parallel to a shift of their genomic enrichment from active TEs to sequences likely derived therefrom but no longer recognizable as such, and that the positive selection of these new functions allowed their KZFP mediators to escape the evolutionary flushing of family members that had become obsolete because only involved in repressing now defunct TEs. This model is supported by additional evolutionary evidence such as (i) the highly dynamic and species-specific populations of KZFPs present in the genomes of all tetrapods (Imbeault *et al*, 2017); (ii) signs of an evolutionary arms race with TE sequences displaying escape mutations following the emergence of their controlling KZFP (Thomas & Schneider, 2011; Jacobs *et al*, 2014; Imbeault *et al*, 2017); (iii) the spreading of canonical transcription factor binding sites in vertebrate genomes via the expansion of lineage-restricted TEs (Bourque *et al*, 2008; Schmid & Bucher, 2010; Schmidt *et al*, 2012; Sundaram *et al*, 2014; Grow *et al*, 2015; Chuong *et al*, 2016); and (iv) the prominent role played by KZFPs to promote the genomewide exaptation of these TEs-derived regulatory sequences through the taming of their transcriptional impact during early embryogenesis (Pontis *et al*, 2019).

Overall, the picture emerging from these data is that of the relentless progression, over more than 400 million years, of two tightly linked evolutionary waves mediating the mechanistic turnover of vertebrate transcriptional networks, one constituted by TEs and the other by their controlling KZFPs, the passage of which occasionally left behind *cis*- and *trans*-acting effectors with novel functions that led to their stabilization by durable exaptation.

# Materials and Methods

## Cell culture

Establishment of the 293T cell lines used for this study was previously described (Imbeault *et al*, 2017). 293T and *KAP1* KO HAP1 cells were cultured in DMEM (Thermo Fisher) supplemented with 10% fetal calf serum and 1% penicillin/streptomycin.

## Immunofluorescences

The cell lines were plated on glass coverslips, and KZFP expression was induced for 4 days with 1 μg/ml doxycylcine. Once at 70% confluency, the cells were washed three times with PBS and fixed with ice-cold methanol for 20 min at −20°C. The cells were then washed three times with PBS and blocked with 1% BSA PBS for 30 min. The fixed cells were then incubated with 1/2,000 anti-HA antibody (HA.11, BioLegend, Covance Catalog# MMS-101P) in 1% BSA PBS for 1 h. The samples were washed three times with PBS and incubated with 1/800 Alexa 488-conjugated anti-mouse antibody (EPFL Histology facility) in 1% BSA PBS for 1 h, and in the last 10 min of the incubation, DAPI solution was added to a final concentration of 1/10,000. For HSPA5 IFs, the primary antibody was obtained from Abcam (ab21685) and the secondary Alexa 647-conjugated anti-rabbit antibody (EPFL Histology facility) was diluted to 1/800. The cells were then washed three times with PBS, the coverslips were mounted on slides with Fluoromount (Merck, F4680), and images were acquired on a ZEISS LSM 700 microscope using the 63×/1.40-oil immersion objective.

## Affinity purification and peptide preparation

Each biological replicate consisted of 100 million HA-tagged KZFP overexpressing 293Ts. Prior to affinity purification, KZFP expression was induced with 1 μg/ml of doxycycline for 4 days. Once confluent, the cells were harvested in PBS 1 mM EDTA. The samples were lysed in HNN lysis buffer (0.5% NP40, 50 mM Tris–HCl, pH 8.0, 150 mM NaCl, 50 mM NaF, 1.5 mM NaVO$_3$, 1 mM EDTA, supplemented with 1 mM PMSF, and protease inhibitors: Sigma P8340) and fixed with 3 mM DSP for 40 min, and reactive DSP was then quenched with 100 mM Tris. The lysates were subjected to 250 U/ml benzonase (Merck, 71205) treatment for 30 min at 37°C. The lysate was then centrifuged for 15 min at 17,000 rcf in order to remove insoluble material. The supernatant was then incubated with 200 μl of pre-washed anti-HA agarose beads (Sigma, ref A2095) for 2 h on a rotating wheel at 4°C. Immunoprecipitates were washed three times with 2 ml and twice 1 ml HNN lysis buffer and three times with 2 ml and twice 1 ml HNN buffer (0.5% NP40, 50 mM Tris–HCl, pH 8.0, 150 mM NaCl, 50 mM NaF). The proteins were then eluted with 3 × 150 μl of 0.2 M glycine, pH 2.5. The samples were then neutralized and denatured with 550 μl 0.5 M NH$_4$HCO$_3$, pH 8.8, 6M urea, reduced with 5 mM TCEP for 20 min at 37°C and alkylated with 10 mM iodoacetic acid for 20 min at room temperature in the dark. The urea concentration was diluted to 1.5 M with 50 mM NH4HCO3 solution. The samples were then digested with 1 ug trypsin (Promega, V5113) overnight at 37°C in the dark. The next day, the digestion was stopped by lowering the pH with the addition of 50 μl of formic acid (FA, AppliChem, A3858.0500) and the peptides were purified using C18 MicroSpin columns (Harvard Apparatus, SEM SS18V) according to the protocol of the manufacturer. They were eluted with 2 × 150 μl 50% acetonitrile, 0.1% FA. The peptides were then dried using a SpeedVac centrifuge.

## Mass spectrometry

Peptides were resuspended in 2% ACN 0.1% FA spiked with iRT peptides at a ratio of 1:20 (Biognosys AG), and ca. 200–400 ng of peptides (20% of the AP sample) was subjected to LC/MS-MS analysis on a LTQ Orbitrap XL mass spectrometer (Thermo Fisher) in two technical replicates. Peptide separation was carried out with a Proxeon EASY-nLC II liquid chromatography system equipped with a RP-HPLC column with emitter (75 μm × 10 cm, New Objective) self-packed with Magic C-18 AQ (3 μm) resin (Bischoff Chromatography). Peptides were separated a linear gradient of solvent B (0.1% formic acid, 98% acetonitrile) in solvent A (0.1% formic acid, 2% acetonitrile) from 5–33% in 90 min, followed by increase to and hold at 80% for 5 min each and at a flow rate of 300 nl/min. From a high-resolution MS1 survey in the Orbitrap (60,000 at 400 *m/z*), the ten highest intense ion signals in the range from 300 to 1,600 *m/z* were isolated (width = 2.5 *m/z*) for fragmentation by CID (NCE = 35.0) and recording of fragment ion spectra in the linear trap quadrupole (LTQ). Singly charged ions were rejected. Only precursor ions with 150 or more ion counts were considered. Dynamic exclusion of up to 300 signals was set to 20 s, allowing for repeat sequencing of the same ion eluting to extend the dynamic range of spectral counting. Scans were performed after reaching the AGC gain targets of 10$^6$ in maximally 500 ms fill time (MS1) and 10$^4$ ions in maximally 100 ms fill time (MS2). In-between AP samples, bovine serum albumin tryptic digest (200 fmol on column) was analyzed in 20-min gradients to monitor LC and MS performance and to minimize peptide carry-over to the next sample. Technical replicates "a" were acquired for all samples before acquiring technical replicates "b", with 10–24 measurements in-between technical replicate analyses. For internal quality control, the iRT peptides (Biognosys AG) were spiked into all samples and included in the database for spectrum identification.

## Protein identification and quantification

Overall, 2 biological replicates of the 101 KZFPs were processed in 29 MS batches, using each time non-transduced and GFP-HA-expressing 293T cells as negative controls with 2 technical replicates analyzed for each sample to reach a total of 520 LC-MS/MS analyses. The acquired spectra were searched against the human UniProt database (reviewed, canonical entries, downloaded 05-2015, http://www.uniprot.org/), supplemented with sequences of the affinity tag construct, affinity matrix polypeptide, a control bait (green fluorescent protein), the iRT peptides (Biognosys), and common LCMS contaminants (http://www.thegpm.org/crap/), and reversed decoy entries (TPP subsetdb –R). Four search engines were employed in parallel, specifically X! TANDEM Jackhammer TPP (2013.06.15.1—LabKey, Insilicos, ISB), OMSSA (omssacl: 2.1.9), MyriMatch (2.1.138), and Comet (version "2016.01 rev. 3"). The analysis was performed in the swiss grid proteomics portal (Kunszt *et al*, 2015). The enzyme was set to trypsin, allowing 2 missed cleavages. Included were "carbamidomethyl (C)" as static and "phospho (STY)

and oxidation (M)" as variable modifications. The mass tolerances were set to 25 ppm for precursor ions and 0.5 Da for fragment ions. The identified peptides were processed and analyzed through the Trans-Proteomic Pipeline (TPP v4.7 POLAR VORTEX rev 0, Build 201403121010) using PeptideProphet, iProphet, and ProteinProphet scoring. The peptide-spectrum matches reported by the four search engines were integrated via iProphet. Peptides propagated to ProteinProphet were filtered at an FDR of 0.01 (iprophet-pepFDR). The assigned proteins were further filtered at a protein level FDR of 0.01 (ProteinProphet) and quantified based on the number of spectra matched to unique, proteotypic peptides.

## AP-MS analysis

All samples selected for further analyses had to display more than 10 bait KZFP spectral count in both technical replicates in order to ensure proper bait protein levels. Only proteotypic, unique spectral counts were used. Fold changes between interactors and controls were computed using the CRAPome website (Mellacheruvu *et al*, 2013). Significance between bait–prey interactions was inferred using the SAINT (Choi *et al*, 2011) software v 0.03 using all controls and all baits to estimate the SAINT probabilistic model as advised in the SAINT manual. Significant interactions were defined as having a BFDR (as defined by SAINT) lower than 0.01 and an average enrichment over control bigger than 2. Using our IF experiments, we classified the KZFPs into 2 pattern categories: nuclear or mixed nuclear/cytoplasmic. We then retrieved the Gene Ontology Cellular Component term identifier ("GO id") of all detected preys through the publicly available Human Protein Atlas dataset (Thul *et al*, 2017) and discarded entities brought down by nuclear-only KZFPs but classified as exclusively cytoplasmic by GO id. Proteins that remained after application of these three filters were considered as *bona fide* KZFP-binding factors. Quantitative measure of KAP1 enrichment, KAP1FC, is the logarithmic (base 2), spectral count fold change of KAP1 compared to controls. A KZFP interactome displaying a KAP1FC below 3 was not considered KAP1 associated (Figs 3A and EV1D). Individual KZFP interactomes were also represented by volcano plots displaying the BFDR and the average fold change for each prey with names indicated for all preys exhibiting a BFDR of 0.1 and fold change above 2.

## Network

The KRAB interactome network is a force-directed network built using Gephi software with the ForceAtlas2 graph layout algorithm (Bastian *et al*, 2009; Jacomy *et al*, 2014). The topology of the network is established by a force-directed process that follows certain rules: All nodes repel each other and are attracted to the center by artificial "gravity", and nodes with links attract each other. The distance of a link between a bait KZFP and its prey is inversely proportional to the measured fold change over controls.

## ChIP experiments

The chromatin was prepared as described in our previous study (Imbeault *et al*, 2017). Quality control of the chromatin was performed on a Bioanalyzer 2100 (Agilent) to verify that most fragments were between 200 and 600 bp. ChIP was performed using 15 μg anti-HA.11 antibody (clone 16B12, Covance and protein G dynabeads) (Thermo Fisher). For the ChIP-exo (Table EV3), the samples were processed as described in our previous study (Imbeault *et al*, 2017), and the libraries were pooled and sequenced 12 per lane on an Illumina HiSeq 2500 to a minimum depth of around 15 million 100 bp single-end reads. For the ChIP-seq experiments (Table EV3), libraries were ligated with Illumina adaptors, and samples were pooled and sequenced on the same lane on an NextSeq 500 to a minimum depth of around 27 million 85 bp single-end reads. Reads were mapped to the human genome assembly hg19 using Bowtie2 v2.3.3.1 short read aligner (Langmead & Salzberg, 2012), using the –sensitive-local mode for ChIP-seq and –very-sensitive-local mode for ChIP-exo. ChIP-seq mapped reads were filtered for mapping quality > 10, and peaks were called using MACS v1.4.2.1 (Zhang *et al*, 2008) with defaults parameters. For each ChIP-seq sample, a corresponding total input sample was set as control. For ChIP-exo, MACS was also used with control as in our previous study (Imbeault *et al*, 2017) and with parameter –keep-dup all. For the rest of the analysis, only peaks with MACS score > 80 were kept.

## Intersections, alignments, graphic representations of data, phylogenetic and conservation analyses

### Bedfile intersections

Intersections and Fisher's exact test were performed using the bedtools suite version 2.27.1 (Quinlan & Hall, 2010). The genomic overlapping KZFPs (Fig EV1B) were determined by intersecting KZFP bedfiles corresponding to their ChIP-seq experiments performed in our previous study (Imbeault *et al*, 2017). All KZFPs sharing 10% of their binding sites in a reciprocal fashion were termed "overlapping". GTF3C2 bedfile was obtained from Encode (ENCFF002CYL), and CTCF bedfile was obtained from Encode version 3. The TSS track corresponds to Ensembl annotated TSSs ± 2,500 base pairs downloaded from UCSC on the 26/08/2014, and the TE track corresponds to DNA transposons and EREs as annotated by RepeatMasker on the 31/01/2014 with LTRs merged as in a previous study (Pontis *et al*, 2019).

### Boxplots

Boxplots are shown as median, and 25th (Q1) and 75th (Q3) percentiles. The upper whisker extends to the last data point less than Q3 + 1.5*IQR, where IQR = Q3−Q1. Similarly, the lower whisker extends to the first data point greater than Q1 − 1.5*IQR.

### dn/ds ratio

We retrieved all available KRAB A-box DNA coding sequences from 69 species ranging from sauropsids to *Homo sapiens* and aligned the KRAB sequences of human KZFPs with that of their evolutionary most distant ortholog(s) using the Clustal Omega suite (Sievers *et al*, 2011). Non-synonymous over synonymous mutation ratios (dn/ds) were then calculated using the BioPython module codonseq cal_dn_ds command applying default parameters. For each human KRAB domain, the dn/ds ratio was calculated based on the dual alignment of the human sequence and the KRAB domain belonging to the KZFP most distant species ortholog. If several species presenting ortholog KRAB domains were equally distant from the human KRAB, average of all dn/ds ratios was given to the KRAB.

### KRAB A-boxes phylogenetic and alignment analyses

Human KRAB A-boxes sequences were extracted from 346 reviewed human KZFPs obtained from the UniProt protein amino acid sequences database that were also detected in the human genome in our previous study (Imbeault *et al*, 2017). Coelacanth KRAB A-boxes sequences were extracted from the 12 annotated KZFPs in the coelacanth proteome (UniProt). Phylogenetic trees and sequence alignments were obtained using the alignment website Clustal Omega using default parameters (McWilliam *et al*, 2013), and the phylogenetic tree was built following default parameters (neighbor-joining tree without distance corrections). The amino acid sequence color pattern used takes into account their biochemical features (http://www.jalview.org/help/html/colourSchemes/zappo.html), and the conservation visibility was set to 30.

### Consensus sequences logos

Sequence alignments performed with Clustal Omega were processed through the Skylign software (Wheeler *et al*, 2014) to generate consensus logos. Human and Coelacanth KRAB consensus logos were generated with these parameters: remove mostly empty columns, alignments sequences are full length, information content —above background. PRDM9s unique sequences were generated with these parameters: use observed counts, alignments sequences are full length, information content—All.

### Heatmaps

Expression of KZFPs in 293T cells was monitored via an in-house RNA-sequencing experiment. This expression was represented in a single-column heatmap obtained using the heatmap function in the python module seaborn v0.9.0. KZFPs expression in human tissues was retrieved from the tissue-specific gene expression GTEx database (The Gtex consortium, 2013). The logged median RPKM value for each bait KZFP in different tissues was represented in a cluster map (Z-score) obtained using the clustermap function in the python module seaborn. Correlation analyses and heatmap visualizations were performed in the R statistical computing environment extended with the pheatmap package.

### KZFP LoF methods

Human genetic exome and whole genome sequencing data were obtained from gnomAD77 (release-2.0.2) for 123,136 and 15,496 individuals, respectively (Lek *et al*, 2016). To ensure a high-quality dataset, the genetic data were further processed and filtered through several steps. First, all single nucleotide variants (SNVs) +/− 1 kb around the KZFP canonical transcripts as defined by Ensembl (v75, hg19) were extracted and filtered for variant quality, thus retaining only variants annotated as "PASS". Second, all indels were normalized and multiallelic variants split using BCFTools (v 1.8) and reannotated with the Variant Effect Predictor 78 and LOFTEE. Third, all loss-of-function (LoF) variants associated with well-covered canonical transcripts were extracted and merged from the exome and whole genome datasets and either low confidence or flagged LoF variants were removed. The latter was primarily due to the LoF variant being found in the last 5% of the canonical transcript. Well-covered transcripts were defined as transcripts having an average per-base coverage > 20×. Furthermore, exons with an average per-base coverage < 20× were also removed. Finally, the number of LoF variants per gene was normalized by the length of the coding sequence and the converted into a Z-score for comparison of constraint between the KZFPs.

### Plasmids and transfection

#### Plasmids

Indicated constructs (Table EV4) were cloned in pDONR221, without STOP codon, and shuttled either in destination vector pTRE-3HA, pTRE-2FLAG (ref Addgene: 631012), or pcDNA6.2 C-eGFP (ref Thermo Fisher: V35520) which produce, respectively, proteins with C-terminal HA, FLAG (in a doxycycline-dependent manner), or GFP tag (in a constitutive manner).

#### Transfection

$7.5.10^6$ cells were transfected with 12.5 μg of plasmid(s) using FuGENE6 (Promega) transfection reagent, according to the manufacturer's protocol. For HA- and FLAG-tagged constructs, doxycycline was added at a concentration of 1 μg/ml for 48 h. 2 days after the transfection, the cells were harvested.

### Immunoprecipitations

$10.10^6$ cells were harvested, washed with PBS, and resuspended in lysis buffer (400 mM NaCl, 10 mM HEPES, pH 7.5, 0.1% NP-40, Sigma P8340); the NaCl concentration was then brought down to 150 mM; and samples were sonicated using a probe sonicator and then centrifuged at 17,000 rcf in order to remove insoluble material. The protein concentration of the lysates was measured using a BCA assay (Thermo Fisher, 23225), and equivalent protein quantities were then incubated with 50 μl of pre-washed anti-HA agarose beads (Sigma, A2095) overnight on a rotating wheel at 4°C. The samples were then incubated five times for 10 min with 1 ml of wash buffer (150 mM NaCl, 10 mM HEPES, pH 7.9, 0.1% NP-40, protease inhibitors) on a rotating wheel at 4°C. Immunoprecipitates were eluted in Laemmli buffer at 95°C for 10 min. Similar elution volumes and protein quantities, for the IPs and the inputs, respectively, were loaded on a gel. Western blots were performed using anti-Sirt1 (Abcam, ab32441), anti-IPO7 (Abcam, ab99273), anti-GFP (Abcam, ab5450), anti-KAP1 (Merck, MAB3662), HRP-conjugated anti-Flag antibody (Sigma, A8592), HRP-conjugated anti-HA antibody (Sigma, 12013819001), HRP-conjugated anti-goat antibody (Dakocytomation, P0449), HRP-conjugated anti-rabbit antibody (Santa Cruz, sc-2004), and HRP-conjugated anti-mouse antibody (GE Healthcare, NA931V).

### Illustrations

Illustrations were obtained from www.somersault1824.com (Fig EV1A) and https://en.silhouette-ac.com (Fig 5C).

## Data availability

The mass spectrometry proteomics data have been deposited to the ProteomeXchange Consortium via the PRIDE partner repository with the dataset identifier PXD011322 (https://www.ebi.ac.uk/pride/archive/projects/PXD011322).

IF experiments have been deposited on Figshare with the accession https://doi.org/10.6084/m9.figshare.8342486. Volcano plots

have been deposited on Figshare with the accession https://doi.org/10.6084/m9.figshare.8342435. The ChIP-seq experiments have been deposited on GEO with the accession number GSE120539.

**Expanded View** for this article is available online.

## Acknowledgements

This work was supported by ERC grants "Proteomics 4D" (ERC-2014-AdG 670821) to RA and "KRABnKAP" (ERC-2011-AdG 268721) and "TRANSPOS-X" (ERC-2016-AdG 694658) to DT and by subsidies from the Swiss National Science Foundation (310030_152879 and 310030B_173337) to DT.

## Author contributions

MH and P-YH designed the project and executed it. P-YH performed affinity purification and ensuing protein digestion as well as peptide purification on KZFPs-overexpressing 293T cell lines, and MH conducted the mass spectrometry analysis of the samples as well as the identifications of the spectral counts. JD set the statistical methods used to calculate the spectral count enrichments. P-YH executed immunofluorescence experiments. CWT analyzed the frequencies of loss-of-function mutations for the KZFP transcripts in the human population. CP analyzed the expression of KZFPs in the GTEx dataset. PY-H handled the plasmid constructs cloning and ensuing immunoprecipitation experiments, and the ChIP-seq experiments were done in association with MI. P-YH, MH, PT, and DT wrote the manuscript. AC and JP made valuable intellectual contributions. RA and DT supported this work.

## Conflict of interest

The authors declare that they have no conflict of interest.

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
