## [Review Process File · The EMBO Journal]

The interactome of KRAB zinc-finger proteins reveals the evolutionary history of their functional diversification

Pierre-Yves Helleboid, Moritz Heusel, Julien Duc, Cécile Piot, Christian W. Thorball, Andrea Coluccio, Julien Pontis, Michael Imbeault, Priscilla Turelli, Ruedi Aebersold and Didier Trono

Review timeline:

Submission date:	23rd Nov 2018
Editorial Decision:	8th Jan 2019
Revision received:	6th May 2019
Editorial Decision:	3rd Jun 2019
Revision received:	3rd Jul 2019
Accepted:	10th Jul 2019

Editor: Ieva Gailite

Transaction Report:

1st Editorial Decision

8th Jan 2019

Thank you for submitting your manuscript for consideration by the EMBO Journal. We have now received three referee reports on your manuscript, which are included below for your information.

As you will see from the comments, all reviewers appreciate the relevance of the resource aspect of the presented data and the described identification of KAP1-independent KZFP functions. However, the reviewers also raise a number of issues would need to be addressed before they can support publication here. Based on the overall interest expressed in the reports I would like to invite you to submit a revised version of your manuscript in which you address the comments of all three referees, but particularly focusing on providing further validation of the data at the endogenous level of protein expression, as requested by all referees.

REFeree REPORTS:

Referee #1:

Krüppel-associated box (KRAB) domain containing zinc finger proteins (ZFPs) are a rapidly evolving family of transcriptional regulators, mostly known for their role in the repression of transposable elements (TEs) through the recruitment of KAP1 (TRIM28) and components of the K9me3 machinery. However, due to the diversity of ZFP binding domains, the function of many of these proteins has not been determined. One possibility to identify roles of these KZFP would be to look at their interacting partners. Here Hellboid et.al., take an affinity-purification linked to Mass-Spectrometry approach to identify the interacting partners of 101 human KZFPs from a range of evolutionary ages. This is an impressive data set with matching IF of the factors revealing also their subcellular localization. They verify that the majority of these KZFPs interact with KAP1 and components associated with TE repression, but interestingly they also identify that a subset of these KZFPs interact with proteins with other functions, including mediators of nuclear import, RNA processing and eukaryotic translation-initiation factors. They also find a group of KZFP they call

variant-KZFPs that are evolutionarily the oldest and interact less with KAP1. In addition they verify some interactions between KZFPs.

There are a few points of concern about the conclusions drawn, but overall this paper's key contribution is that it provides highly relevant data as a resource. My major concern is that the authors seem to overinterpret in regards to what they learned about the evolution and function by just looking at interaction partners. These statements would need to be toned down and/or supported by additional evidence, which should make this manuscript highly suitable for publication in EMBO Journal.

Major points of concern:

1) One of the main conclusions of the paper is that some KZFPs have evolved additional functions aside from TE repression. However, because the experiments were performed in an overexpression system it is hard to know how significant these results are. Some of the differences in interacting partners could be the result of variable and/or high expression levels of the different KZFPs. Particularly because the interactors are abundant proteins in the cell (e.g. RNA processing factors, translation initiation factors) and therefore could be non-specific interactors or contaminants. It would be nice if some of these interactions could be validated and shown to be important in a process other than TE repression.

They argue that once the TEs that used to be suppressed by these KZFP become inactive, the repressive role of the KZFPs was no longer needed and that the loci now occupied by these KZFPs have different functions. However, the one example they give is of a LINE1 element that can no longer transpose but is still bound by KZFPs that now interact with RNA processing factors. But again it is hard to know if these interactors are specifically needed for some newly evolved function, are still remaining from when they were involved in repressing LINE1 elements or is just a feature of that locus that does not have any specific function.

2) The identification of standard-KZFPs and variant-KZFPs that are strong and weak KAP1 binders respectively and that the vKZFPs are evolutionarily older and interact with more non-canonical partners or with fewer partners is also interesting. However, I am not sure it validates the statement that this is support for the evolution of non-canonical or cell-restricted biological roles. I think there is a lack of support for this being of functional relevance and not just that the vKZFPs have been evolving for longer and therefore are more divergent from their younger sKZFPs relatives, having just weakened and lost interacting partners or now interact with variable partners that do not serve a significant biological function.

3) Their identification and validation of the potential role of the DUF3669 domain in the oligomerization of proteins is interesting. However, the KZFPs were overexpressed in cells and the IPs performed in cell lysates, which does not preclude the possibility that the interactions are indirect. To make this conclusion the proteins would need to be purified and tested for an interaction *in vitro*. This statement should be toned down.

4) Currently the discussion is highly speculative and would benefit from a paragraph summarizing the value of having comprehensive interaction maps that also discusses the potential limitation of an overexpression system

Minor points of concern:

There are some grammatical errors throughout the text and in the figure legends.

Line 129: was, should be were.

Lines 161-162, 174, 247, 249: Numbers below 10 should be written.

Referee #2:

In this manuscript, Helleboid et al. characterized the interactome of 101 Krüppel-associated box (KRAB)-containing zinc-finger proteins (KZFPs). Using KZFPs as baits for affinity purification and mass spectrometry, they identified 219 KZFP binding proteins in 293T cells after quality filtration. As expected, the KZFP interactome global network contained Kap1 at its core. Beyond that, the

authors identified sub-networks consisting of KZFPs harboring SCAN and DUF domains, and demonstrated their oligomerization potential using protein co-immunoprecipitation. Next, they focused on KZFPs with multiple unique interactors to provide more detailed insight into possible biological roles. Subsequently, the authors explored the determinants of KZFPs affinity towards Kap1 binding, by relating that affinity to conservation of KRAB-A box. Phylogenetic analysis showed that KZFPs in which KRAB A-box has significant divergence from the consensus sequence (vK-KZFPs), interacted less strongly with KAP1 compared to other KZFPs in which KRAB A-box is more similar to the consensus sequence (sK-KZFPs). Comparison of evolutionary ages between vK-KZFPs and sK-KZFPs (based on results from Imbeault et al, 2017), showed vK-KZFPs are significantly older than sK-KZFPs. Finally, the authors tested the Kap1 binding potential for KZFPs in the last common ancestor of coelacanth and tetrapods, and contrary to their expectations, found that coelacanth-KAP1 binds to coelacanth-KRAB domains, but not to human ZNF93 nor coelacanth or human PRDM9. Based on these nicely performed experiments, the authors propose a model in which KAP1 recruitment was a relatively early ancestral property of KZFPs, therefore KZFPs first emerged as TE-controlling repressors.

This work by Helleboid et al constitutes important contribution for KZFPs and transposable elements research in two aspects. First, it provides extensive reference for protein-protein interactions of human KZFPs, especially if combined with previously reported protein-DNA interactions (Imbeault et al, 2017). Second, this work provides more support for a prevailing model in which KZFPs emerged primarily as Kap1 recruiting factors to suppress TEs and TE born enhancers. Still, there are some experimental limitations which should be discussed more thoroughly to improve the manuscript.

1) One limitation of the study is that the interactome of KZFPs is based on proteins exogenously expressed in 293T cells. While this is inevitable for such a large-scale analysis, it should be clearly pointed out by the authors as a limitation of their study. This is of special importance for KZFPs, as they have unique expression patterns during development in different tissues, leading to under estimation of KZFP-KZFP interactions, as well as of KZFPs interaction with preys that has unique tissue distribution timed with their baits. For example, ZNF263, ZNF 277, and ZNF212 are highly expressed in germ cells and many/some of their true interacting partners may not be expressed in 293T. What is the fraction of KZFPs used in this study are endogenously expressed in 293T cells? Was endogenous expression a factor considered in selection of representative candidates? Also, adding supplementary data showing endogenous expression will be valuable as indicator of likelihood of false negative results in the case of KZFPs that are not endogenously expressed. If one KZFPs is endogenously expressed, it is more likely its interacting proteins are co-expressed, and vice-versa.

2) For the SCAN domain containing KZFPs, are there any unique features of their interactome? Do they share common preys? The authors didn't elaborate in more details other than their oligomerization potential.

3) In figure 3.B: adding one more column showing the domain architecture of the protein will be informative for the reader.

4) In figure 3.D: It would be interesting to know the correlation between percentage of peaks in TEs and the binding affinity to Kap1 (if bound). One would expect a positive correlation between them as stronger binding to Kap1 seems critical for TE suppression. But It will be interesting also if binding to TEs is not associated with Kap1 binding affinity, indicative of some roles of Kap1 other than TE repression.

Minor Issues:

1) In Figure 2.C: the scheme for the ZNF282 construct is confusing. It is better to show two constructs (separate DUF and KRAB constructs)

2) In Figure 3.C: is this generated from public datasets? Please refer to the source in the legend if so.

Referee #3:

In this paper, Helleboid et al. present a comprehensive interactome of 101 KRAB zinc finger proteins (KZPFs) in 293T cells. They also perform ChIP-seq and ChIP-exo sequencing to investigate genome wide binding patterns of the KZPFs and perform evolutionary analyses. In general, the data presented in this paper are of good quality, albeit rather descriptive, and of interest to the scientific community. However, there are some major technical concerns that the authors should address prior to publication

Major points

- The authors should at least discuss or mention the fact that their study makes use of a tagging strategy that results in overexpression of baits. Thus, some of their detected interactions or other data may be the result of overexpression. Given these facts, the authors also need to verify some of their observations at the endogenous level (see below)
- According to their material and methods section (page 14), two affinity purifications were performed for each tagged bait, and each of these was measured twice (technical replicates) by LC-MS. Negative control IPs were also performed. This means that the authors have the ability to plot individual KRAB zinc finger protein (KZFPs) pull-downs against negative control pull-downs, for example in the form of volcano plots, or a larger analysis of multiple or all tagged baits against negative controls in a hierarchical clustering analysis. Such data visualisation is essential to judge the quality of their interactome. Such data visualisation should complement their 'hairball' interactome shown in Figure 1.
- The authors perform various validation IPs using western blotting as a readout (i.e. Figure 5B, Figure S1E), but all of these are done with tagged proteins which are overexpressed using transfection. It would be important to verify at least one or two detected interactions by co-IPs using antibodies against endogenous proteins.
- Particularly given the fact that the authors make use of an AP-MS strategy which results in overexpression of baits, it is essential to verify some of the detected interactions at the endogenous level.
- On a similar note, ChIP-seq and ChIP-exo was also performed using overexpression lines. Again, validation of one or two KZPFs using endogenous antibodies in ChIP-seq or ChIP-exo experiments is very important.
- ChIP-seq analyses can be improved. For example, is there any overlap with histone marks or DNA methylation? Which genomic elements are enriched apart from TEs? Etc etc. The authors could and should be able to retrieve much more information from their data and visualize this data in a nicer way. Figures in general do not look very pretty and informative. The same holds true for their AP-MS datasets.
- The authors use benzonase during lysate preparation, which should efficiently digest nucleic acids. Still, it is important to verify that long DNA fragments are not present in their lysates, since these could be a source of indirect, DNA mediated interactions. Since KZFPs are DNA binding proteins, such controls are very important, particularly since IPs are washed with physiological salt levels.

Minor points

- Some of the figures are difficult to read (i.e. Fig S3C) due to very small font sizes

Dear editor,

We thank the reviewers for their constructive and helpful comments and you for your editorial work. We hope that with these additions and modifications our work will now be found suitable for publication in the EMBO journal.

Changes brought to the manuscript

The referees' comments lead us to make the following changes, explained in more detail in our response to reviewers:

Figures added:

Figure 5: Responses Lines 257-260 (Manuscript Lines 265-278)

Figure S1C: Responses Lines 46-61 (Manuscript Line 126-130)

Figure S2: Responses Lines 257-260 (Manuscript Line 132)

Figure S8: Responses Lines 156-160 (Manuscript Lines 245-252)

Figure S9: Responses Lines 109-110 (Manuscript Lines 260-262)

Volcano plots available on our server: Responses Lines 205-212 (Manuscript Lines 473-475)

Figures modified:

Figure 2C: Responses Line 177

Figure 3B: Responses Line 172

Figure S3C (New name: Figure S6): Responses Line 271

Changes brought to the text in response to comments:

Responses Lines 63-69

Responses Lines 89-93

Responses Lines 170-171

Responses to reviewers

Referee #1

However, because the experiments were performed in an overexpression system it is hard to know how significant these results are. Some of the differences in interacting partners could be the result of variable and/or high expression levels of the different KZFPs.

In order to answer this concern we present here the Western blots performed on all the HA-tagged KZFP-overexpressing 293T cell lines for our previous study (Figure for reviewers - 1). It will allow the reviewers to verify that the expressions of KZFPs with "classical" and "unusual" interactomes do not significantly differ.

Particularly because the interactors are abundant proteins in the cell (e.g. RNA processing factors, translation initiation factors) and therefore could be non-specific interactors or contaminants. It would be nice if some of these interactions could be validated and shown to be important in a process other than TE repression.

We reckon that it would have been nice to perform the same studies on endogenous native rather than exogenously expressed tagged proteins, and even better in cells in which the proteins in question were naturally expressed. However, this would have precluded any large-scale screen, not only because it would have required using a wide array of different cells, but also because antibodies against most KZFPs are simply not available, at least to perform the type of experiments described here. Still, as a gesture of good faith, and even though we were not convinced that verifying a few interactions would do much to validate the 887 detected in our study, we purchased antibodies against ZNF2, ZNF20, ZNF79, ZKSCAN4, ZNF597, ZNF746, ZNF202, ZNF3, ZNF764. Unfortunately, 7 of these 9 reagents were not specific enough for further experiments, and the other 2 not suitable for immunoprecipitations as one did not bind efficiently to the protein G-coupled beads and the other could not immunoprecipitate its target. After several weeks of futile efforts, we thus decided that, instead, we would explain better in the text why our results were still worthy of consideration and further compared our data to the Biogrid Protein-Protein Interaction (PPI) database (Stark, 2005) (Suppl. Table 2), finding that 88 (about 8%) of the interactions documented in our system had been previously detected either through AP/MS or other approaches such as two-hybrid screens (Fig. S1C). Together, these points are now summed up at the beginning of our discussion (Manuscript lines 302-309):

“Owing to the large number of tested baits and the lack of antibodies allowing for an efficient purification of their endogenous versions, we relied on the overexpression of tagged proteins in 293T cells. However, in addition to the performance of biological replicates, the use SAINT probabilistic scoring, batch-specific fold-change-over-control normalization and subcellular localization-based filters allowed us to establish a high-confidence list of interactors. This was supported by our verification that a substantial subset of the interactions detected through our approach had been previously documented in different experimental settings.”

Finally, the interactions detected with factors not related to KAP1-mediated TE repression may not have been yet shown to be functionally important, but several publications have noted the importance of variant KZFPs (ZNF746 and ZNF473) and their interactors in processes such as gene regulation, disease progression and RNA metabolism (Kang & Shin, 2015; Azzouz *et al*, 2005; Shin *et al*, 2011).

They argue that once the TEs that used to be suppressed by these KZFP become inactive, the repressive role of the KZFPs was no longer needed and that the loci now occupied by these KZFPs have different functions. However, the one example they give is of a LINE1 element that can no longer transpose but is still bound by KZFPs that now interact with RNA processing factors. But again it is hard to know if these interactors are specifically needed for some newly evolved function, are still remaining from when they were involved in repressing LINE1 elements or is just a feature of that locus that does not have any specific function.

It seems to us unlikely that i) a KZFP, ii) its target sequence and iii) their interaction would be maintained over 25-30 million years of evolution without it having any functional relevance. The species-specificity of TE-based enhancers that are key to embryonic genome activation supports our assertion, since these “teenhancers” completely differ in macaque and human, two species merely 30 million years apart (Pontis *et al.*, 2019). Still, we amended our text as follows (lines 198-204): “The presence of escape mutations in the LINE1 lineage (Imbeault *et al*, 2017; Jacobs *et al*, 2014) indicates that ZNF93, ZNF765 and ZNF248 initially acted as *bona fide* inhibitors of transposition. Yet their persistent association with L1PA5/PA6 integrants suggests that these KZFPs, while perhaps still repressing transcription, do so no longer to block retrotransposition, since their target retroelements have long lost all spreading potential.”

2) The identification of standard-KZFPs and variant-KZFPs that are strong and weak KAP1 binders respectively and that the vKZFPs are evolutionarily older and interact with more non-canonical partners or with fewer partners is also interesting. However, I am not sure it validates the statement that this is support for the evolution of non-canonical or cell-restrict biological roles. I think there is a lack of support for this being of functional relevance and not just that the vKZFPs have been evolving for longer and therefore are more divergent from their younger sKZFPs relatives, having just weakened and lost interacting partners or now interact with variable partners that do not serve a significant biological function.

We agree with the reviewer that evidences for cell-restricted roles are too thin and we removed this overly speculative comment from the text. However, older KZFPs display the vK domain that is not capable to recruit KAP1. KAP1-mediated silencing of TEs is considered the canonical function of KZFPs. So, by definition, the older KZFPs have to fulfill non-canonical functions. This is supported by the strong conservation of their KRAB domains and DNA fingerprints over long periods of time. Additionally, the level of polymorphism for vKZFPs is lower than their sK counterparts in the human population supporting further the importance of their role (Fig. S9). Our work gives an insight into the potential functions of some vKZFPs endowed with unusual interactors. However, a functional validation of these proteomic results is beyond the scope of this manuscript. Of note, some of these vKZFPs functions have already been investigated, through studies that pointed to broad and diverse functions (Chauhan *et al*, 2013; Shin *et al*, 2011; Azzouz *et al*, 2005; Wagner *et al*, 2000). Furthermore, a number of KZFPs displaying unique interactors were younger and were strong KAP1 recruiters (i.e. ZNF765), suggesting that non-canonical functions are not restricted to older KZFPs and therefore that the time for which the KRAB has been around cannot be the sole predictor of a non-canonical function.

3) Their identification and validation of the potential role of the DUF3669 domain in the oligomerization of proteins is interesting. However, the KZFPs were overexpressed in cells and the IPs performed in cell lysates, which does not preclude the possibility that the interactions are indirect. To make this conclusion the proteins would need to be purified and tested for an interaction *in vitro*. This statement should be toned down.

We agree with the referee and toned down our conclusion accordingly (Manuscript lines 159-162).

4) Currently the discussion is highly speculative and would benefit from a paragraph summarizing the value of having comprehensive interaction maps that also discusses the potential limitation of an overexpression system

We added a paragraph discussing such considerations (Manuscript lines 302-309, see above).

Minor points of concern:

There are some grammatical errors throughout the text and in the figure legends.

Line 129: was, should be were.

Lines 161-162, 174, 247, 249: Numbers below 10 should be written.

Referee #2:

1) One limitation of the study is that the interactome of KZFPs is based on proteins exogenously expressed in 293T cells. While this is inevitable for such a large-scale analysis, it should be clearly pointed out by the authors as a limitation of their study. This is of special importance for KZFPs, as they have unique expression patterns during development in different tissues, leading to under estimation of KZFP-KZFP interactions, as well as of KZFPs interaction with preys that has unique tissue distribution timed with their baits. For example, ZNF263, ZNF 277, and ZNF212 are highly expressed in germ cells and many/some of their true interacting partners may not be expressed in 293T. What is the fraction of KZFPs used in this study are endogenously expressed in 293T cells? Was endogenous expression a factor considered in selection of representative candidates? Also, adding supplementary data showing endogenous expression will be valuable as indicator of likelihood of false negative results in the case of KZFPs that are not endogenously expressed. If one KZFPs is endogenously expressed, it is more likely its interacting proteins are co-expressed, and vice-versa.

Upon performing RNA-seq analysis of control 293T cells, we could detect transcripts from 93 out of the 101 KZFPs used in this study (Fig. S8.A). Moreover, the three KZFPs devoid of interactors, that is, ZNF777, ZNF763 or ZNF212, were in the top third of KZFPs expressed in 293T cells. As well, some KZFPs endowed with unique interactomes were highly expressed (ZNF764, ZNF3, ZNF445) in 293T cells whereas others lowly (ZNF20, ZNF93) or not (ZNF2) expressed in these cells.

2) For the SCAN domain containing KZFPs, are there any unique features of their interactome? Do they share common preys? The authors didn't elaborate in more details other than their oligomerization potential.

Apart from ZNF446 that displays an extensive SCAN interactome, other SCAN-KZFPs do not present specific features and do not share non-SCAN additional interactors (Figure 5). We did not elaborate further on the ZNF446 interactome or SCAN-mediated interactions because two studies already discussed this topic (Schmitges *et al*, 2016; Huttlin *et al*, 2015). We specified it in the text (lines 149-151): " These results concur with some of the putative SCAN-mediated interactions noted in previous studies (Schmitges *et al*, 2016; Huttlin *et al*, 2015)".

3) In figure 3.B: adding one more column showing the domain architecture of the protein will be informative for the reader.

We modified Fig. 3.B accordingly.

Minor Issues:

1) In Figure 2.C: the scheme for the ZNF282 construct is confusing. It is better to show two constructs (separate DUF and KRAB constructs)

We modified the figure 2.C accordingly

2) In Figure 3.C: is this generated from public datasets? Please refer to the source in the legend if so.

We added the references of these bed-files in the figure legend.

Referee #3:

- The authors should at least discuss or mention the fact that their study makes use of a tagging strategy that results in overexpression of baits. Thus, some of their detected interactions or other data may be the result of overexpression. Given these facts, the authors also need to verify some of their observation at the endogenous level (see below)

We agree with the referee and added a paragraph discussing the limitations of our approach (line 301-309). Besides, see our response to referee 1.

-According to their material and methods section (page 14), two affinity purifications were performed for each tagged bait, and each of these was measured twice (technical replicates) by LC-MS. Negative control IPs were also performed. This means that the authors have the ability to plot individual KRAB zinc fingers protein (KZFPs) pull-downs against negative control pull-downs, for example in the form of volcano plots, or a larger analysis of multiple or all tagged baits against negative controls in a hierarchical clustering analysis. Such data visualisation is essential to judge the quality of their interactome. Such data visualisation should complement their 'hairball' interactome shown in Figure 1.

In order to address these concerns, we plotted the False Discovery Rate (FDR) and the Fold change over the controls in the form of volcano plots for every KZFP. These plots are available via this link: <https://drive.switch.ch/index.php/s/kZ2iD1asXO2sU98>. These graphs also indicate the unselected preys close below the FDR threshold (between 0.1 and 0.01 FDR). In order to complement the "hairball" interactome we also built a dot plot depicting all the KZFP bait and their preys (Fig. S2). The same type of information emerges from this graph: most KZFPs recruit KAP1 and its associated proteins while a minor fraction associated with unique interactors. In some cases these partners were different subunits of the same functional complex.

- The authors perform various validation IPs using western blotting as a readout (i.e. Figure 5B, Figure S1E), but all of these are done with tagged proteins which are overexpressed using transfection. It would be important to verify at least one or two detected interactions by co-IPs using antibodies against endogenous proteins. Particularly given the fact that the authors make use of an AP-MS strategy which results in overexpression of baits, it is essential to verify some of the detected interactions at the endogenous level.

For the figures illustrating PO7 and SIRT1 validations (Formerly Fig. S1C, S1E, now Fig. S3A, S3C) only the bait KZFP was overexpressed, whereas the preys were endogenous proteins detected with specific antibodies. For bait KZFPs, as discussed above, despite heroic and costly efforts (both in time and reagents), we could not get commercially available KZFP-specific antibodies to work properly. Of note, in this largely uncharted field, antibodies are put on sale by companies but by and large have not yet left any trace in the literature that could serve as a reference. We further respectfully suggest that confirming with endogenous proteins "some" of the 887 interactions detected in our study would go far less distance in validating our approach than the high stringency filters that were applied, and the crossing of our and previously published data. For the DUF-, the coelacanth KRAB- and KAP1-related experiments, DUF-, coelacanth KRAB- and coelacanth KAP1-specific antibodies are not available. Neither is a coelacanth cell line, precluding the analysis of endogenous proteins.

- On a similar note, ChIP-seq and ChIP-exo was also performed using overexpression lines. Again, validation of one or two KZFPs using endogenous antibodies in ChIP-seq or ChIP-exo experiments is very important.

A recent study (Takahashi et al., 2019) performed Chip-qPCR experiments focused on Imprinting Control Regions (ICR) bound by ZNF445 using an endogenous antibody (anti-ZNF445, PA5-52322, ThermoFisher). Similar to our observations using an anti-HA antibody on ZNF445-overexpressing 293T cells (Imbeault et al., 2017) it confirmed the binding of ZNF445 to ICRs. In the past, others and we have also validated results obtained with another KZFP, ZFP57. Yet, the paucity and poor quality of available KZFP-specific antibodies precludes any scaling up of these efforts. And mapping the genomic targets of human KZFPs was the focus of a previous paper of ours (Imbeault et al., 2017), the results of which relied also on matching with the binding of endogenous KAP1 in hESC, the nature of chromatin marks at these loci, the performance of repression assays, etc. We refer the reviewer to this other publication, as the present work is instead centered on the protein interactome of KZFPs.

- ChIP-seq analyses can be improved. For example, is there any overlap with histone marks or DNA methylation? Which genomic elements are enriched apart from TEs? Etc etc. The authors could and

should be able to retrieve much more information from their data and visualize this data in a nicer way. Figures in general do not look very pretty and informative. The same holds true for their AP-MS datasets.

Again, we think that the reviewer's interrogations are addressed in some of our recent publications (e.g. Imbeault et al. 2017; Pontis et al. 2019).

In an attempt to ameliorate the pedagogic value of our data representations we added two new figures (Fig. 5 and Fig. S2), which we think contribute to illustrating the messages of our study. As for esthetics, we were somewhat at a loss (and saddened) to read that our figures "do not look very pretty". Still, we made some changes in Fig. 4 and 6, which we hope will satisfy the reviewer's taste.

- The authors use benzonase during lysate preparation, which should efficiently digest nucleic acids. Still, it is important to verify that long DNA fragments are not present in their lysates, since these could be a source of indirect, DNA mediated interactions. Since KZFPs are DNA binding proteins, such controls are very important, particularly since IPs are washed with physiological salt levels.

We performed gel migration analyses of the DNA contained in our samples, to find that fragments length was almost exclusively below 500 bp (Figure for reviewers - 2). Moreover, using our Chip-seq data, we verified that pairs of KZFPs recruited to the same loci (ChIP-seq peaks overlapping by at least 10% in a reciprocal manner) did not share more interactors than random pairs of KZFPs (Fig. S1B). It indicates that the artefactual, DNA-mediated pull-down of spurious KZFP interactors did not occur.

Minor points

- Some of the figures are difficult to read (i.e Fig S3C) due to very small font sizes

We thank the reviewer for this remark and modified Fig. S3C accordingly (new Fig.S6).

2nd Editorial Decision

6th May 2019

Thank you for submitting a revised version of your manuscript. The manuscript has now been seen by the original referees. While reviewers #1 and #2 find that their main concerns have been addressed, reviewer #3 is not satisfied with the manner in which the issue of verification of interactions at endogenous protein level was addressed. I have consulted with reviewer #2 on this issue, and he/she found the provided explanation for the experimental difficulties in performing the requested experiments reasonable. I agree with this assessment and will be happy to extend formal acceptance of the manuscript once the following editorial issues are addressed

REFeree REPORTS:

Referee #1:

The revised manuscript provides a highly valuable resource for KRAB ZnF biology. Presentation and conclusions are more balanced and improved.

Referee #2:

The authors have addressed my concerns in a reasonable manner. I recommend publication.

Referee #3:

In this revised manuscript, the authors have not properly addressed one of the major criticisms that was made by all three reviewers, namely validations of interactions at the endogenous level. They have apparently tried to perform endogenous co-IPs for the Znf proteins but these have failed due to the absence of appropriate antibodies (I am a bit puzzled about the reason for one such antibody failing: due to its inability to bind to protein G??? (then use protein A beads!). Anyway, it is now of

course also quite straightforward to tag endogenous alleles using CRISPR. I leave it up to the journal to decide whether this dataset can be published without any protein-protein interaction validations at endogenous expression levels.

2nd Revision - authors' response

3rd Jul 2019

The authors performed the requested editorial changes.

3rd Editorial Decision

10th Jul 2019

Thank you for submitting the revised version of your manuscript. The main issues have now been addressed and I am happy to inform you that your manuscript has been accepted for publication. Congratulations on a nice study!

YOU MUST COMPLETE ALL CELLS WITH A PINK BACKGROUND ↓
PLEASE NOTE THAT THIS CHECKLIST WILL BE PUBLISHED ALONGSIDE YOUR PAPER

Corresponding Author Name: Didier Trono
Journal Submitted to: Embo journal
Manuscript Number: EMBOJ-2018-101220R